METHODS AND RESOURCES

# MSP-tracker: A versatile vesicle tracking software tool used to reveal the spatial control of polarized secretion in *Drosophila* epithelial cells

**Jennifer H. Richens**[1ʘ], **Mariia Dmitrieva**[2ʘ], **Helen L. Zenner**[1ʘ], **Nadine Muschalik**[3], **Richard Butler**[4], **Jade Glashauser**[5], **Carolina Camelo**[5], **Stefan Luschnig**[5], **Sean Munro**[3], **Jens Rittscher**[2,6], **Daniel St Johnston**[1]*

1 The Gurdon Institute and the Department of Genetics, University of Cambridge, Cambridge, United Kingdom, 2 Institute of Biomedical Engineering (IBME), Department of Engineering Science and the Big Data Institute, Li Ka Shing Centre for Health Information and Discovery, University of Oxford, Oxford, United Kingdom, 3 MRC-Laboratory of Molecular Biology, Francis Crick Avenue, Cambridge Biomedical Campus, Cambridge, United Kingdom, 4 The Gurdon Institute, University of Cambridge, Cambridge, United Kingdom, 5 Institute of Integrative Cell Biology and Physiology, Cells in Motion (CiM) Interfaculty Centre, University of Münster, Münster, Germany, 6 Ludwig Institute for Cancer Research, Nuffield Department of Clinical Medicine, University of Oxford, Oxford, United Kingdom

ʘ These authors contributed equally to this work.
* d.stjohnston@gurdon.cam.ac.uk

## Abstract

Understanding how specific secretory cargoes are targeted to distinct domains of the plasma membrane in epithelial cells requires analyzing the trafficking of post-Golgi vesicles to their sites of secretion. We used the RUSH (retention using selective hooks) system to synchronously release an apical cargo, Cadherin 99C (Cad99C), and a basolateral cargo, the ECM protein Nidogen, from the endoplasmic reticulum and followed their movements to the plasma membrane. We also developed an interactive vesicle tracking framework, MSP-tracker and viewer, that exploits developments in computer vision and deep learning to determine vesicle trajectories in a noisy environment without the need for extensive training data. MSP-tracker outperformed other tracking software in detecting and tracking post-Golgi vesicles, revealing that Cad99c vesicles predominantly move apically with a mean speed of 1.1 µm/sec. This is reduced to 0.85 µm/sec by a dominant slow dynein mutant, demonstrating that dynein transports Cad99C vesicles to the apical cortex. Furthermore, both the dynein mutant and microtubule depolymerization cause lateral Cad99C secretion. Thus, microtubule organization plays a central role in targeting apical secretion, suggesting that *Drosophila* does not have distinct apical versus basolateral vesicle fusion machinery. Nidogen vesicles undergo planar-polarized transport to the leading edge of follicle cells as they migrate over the ECM, whereas most Collagen is secreted at trailing edges. The follicle cells therefore bias secretion of different ECM components to opposite sides of the cell, revealing that the secretory pathway is more spatially organized than previously thought.

**Data availability statement:** The MSP-tracker and MSP-viewer programmes are available at https://doi.org/10.17863/CAM.114339 along with a training manual and 2 instructional videos. The training dataset is available at https://doi.org/10.17863/CAM.114338.2.

**Funding:** This work was supported by Wellcome Trust Collaborative awards (095927, 203285), a Wellcome Principal Fellowship to DStJ (080007, 207496), a BBSRC project grant (BB/P026486/1) and by centre grant support from the Wellcome Trust (092096, 203144) and Cancer Research UK (A14492, A24823). Work in SL's laboratory was supported by the Deutsche Forschungsgemeinschaft (SFB 1348 "Dynamic Cellular Interfaces") and the "Cells-in-Motion" Cluster of Excellence (EXC 1003-CiM). The funders had no role in study design, data collection and analysis, decision to publish, or preparation of the manuscript.

**Competing interests:** I have read the journal's policy and the authors of this manuscript have no competing interests.

**Abbreviations:** BN, Bayesian network; CNN, convolutional neural network; DL,deep learning; ER,endoplasmic reticulum; MSP,multiscale particle; MSSEF, multiscale spot enhancing filter; ROIs, regions of interest; RUSH, retention using selective hooks; SBP,Streptavidin-binding peptide; SEF,spot-enhancing filter; SNR,signal-to-noise ratios; TGN,trans-Golgi network.

## Introduction

Many tissues and organs are composed of tubes or sheets of polarized epithelial cells that act as barriers between internal compartments or between the inside and outside of the body. A key function of epithelia is to control the passage of ions, solutes, and nutrients from one side of the epithelium to the other, and this requires localization of ion channels and transporters to either the apical or basolateral domains of the plasma membrane and the formation of lateral occluding junctions that act as barriers to paracellular transport [1,2]. In addition, many epithelia secrete specific factors from their apical or basal surfaces. To perform these diverse functions, epithelial cells must target secreted and transmembrane proteins to either the apical or basolateral plasma membrane.

Many secreted and transmembrane proteins are localized to the correct domain by polarized exocytosis. In this direct pathway, newly synthesized proteins traffic through the endoplasmic reticulum (ER) and Golgi to the *trans*-Golgi network (TGN), where they are sorted into post-Golgi carriers that move to and then fuse with the appropriate membrane [3,4]. Other proteins are localized by different mechanisms, however, such as diffusion and trapping or secretion to the opposite membrane, followed by polarized endocytosis and recycling. The direct route can be distinguished from indirect localization mechanisms by following newly synthesized protein molecules on their first passage to the plasma membrane by blocking them in the ER or Golgi and then releasing them. This is possible for specific proteins, such as a temperature-sensitive form of VSV-G protein and *Drosophila* Rhodopsin [5–10]. An alternative, more generally applicable strategy exploits the observation that proteins are retained in the Golgi of mammalian cells in culture at 20°C, allowing their release as a synchronous wave of secretion by raising the temperature [11]. Experiments using these approaches have identified multiple sorting signals for apical and basolateral cargoes and have revealed that there are a variety of trafficking routes to each membrane domain, some of which pass through intermediate endosomal compartments [2,12,13]. Nevertheless, it is still unclear how specific cargoes are targeted to the appropriate membrane domain and how this is controlled by the apical-basal polarity system [1].

Although these approaches have proved powerful, they involve nonphysiological conditions or work only for specific cargoes. For example, the low temperature used to arrest secreted proteins in the Golgi can also affect other cellular processes, while the total block to the exit of proteins and lipids from the TGN might have knock-on effects on the localization of essential trafficking factors. These limitations have been overcome by the development of the Retention using Selective Hooks (RUSH) technique [14]. In this approach, the labeled cargo is fused to the Streptavidin-binding peptide (SBP), and the hook is provided by Streptavidin fused to KDEL, he cargo in the ER, until the addition of biotin triggers its release by binding with high affinity to Streptavidin and displacing the SBP-tagged cargo, which continues along the secretory pathway.

Most studies of secretion in epithelia have been performed in a small number of immortalized mammalian cell lines, such as Madin-Darby canine kidney (MDCK) cells, derived from the kidney tubule epithelium. We therefore set out to exploit RUSH to investigate how secretion is targeted to the apical and basolateral domains in a more typical secretory epithelium that is amenable to genetic analysis, using the *Drosophila* follicular epithelium as a model [15]. Our results highlight the essential role of microtubule-based transport in targeting secretion and demonstrate that epithelial cells target secretion to multiple specific destinations.

## Results

### Using RUSH to visualize polarized secretion

To examine apical secretion using RUSH in the *Drosophila* follicle cells, we inserted SBP and the HaloTag self-labelling tag after the signal peptide of Cadherin 99c (Cad99c) under the control of the GAL4/UAS system (Fig 1B). Cadherin 99c is the *Drosophila* orthologue of the

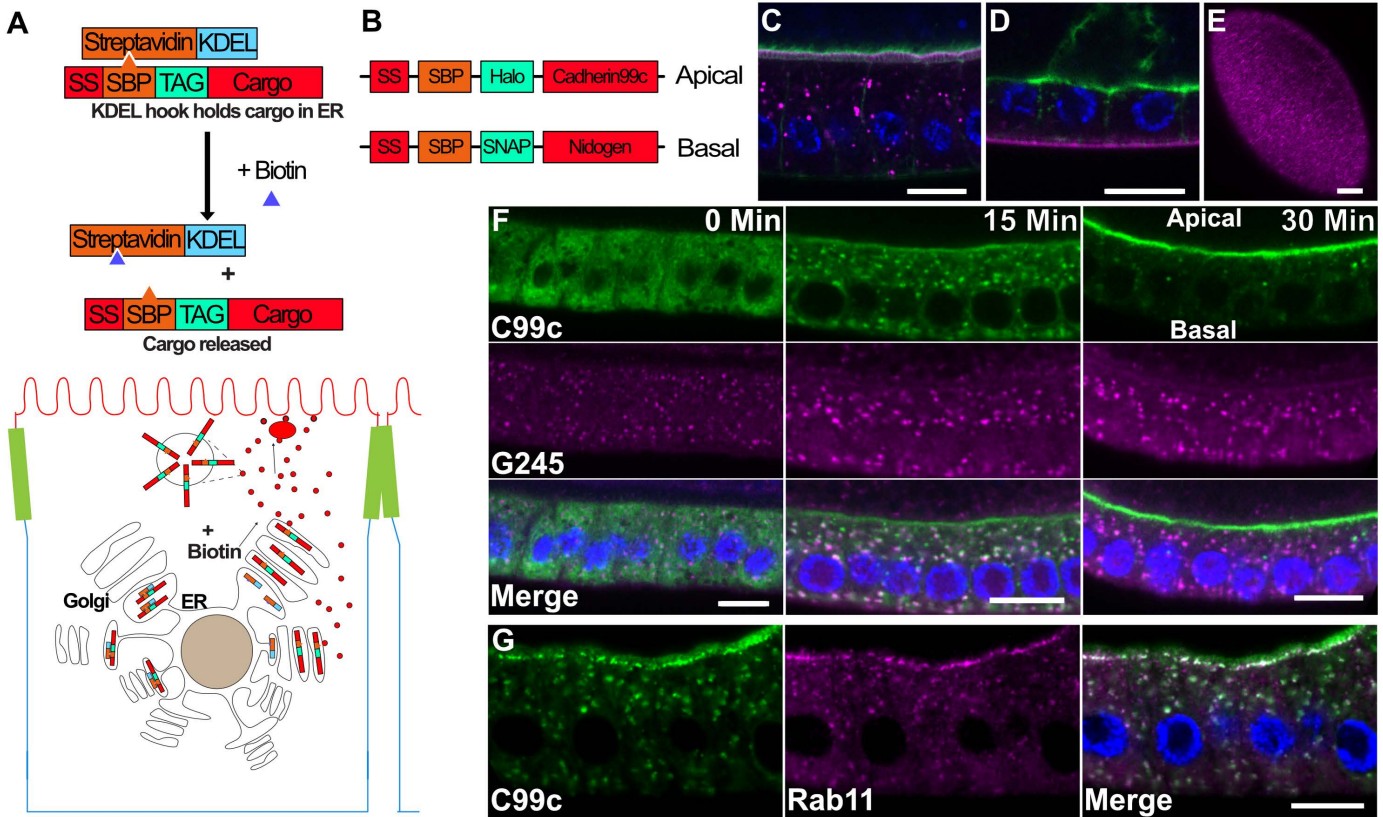

**Fig 1. The RUSH system. A)** Diagram of the RUSH system. The Halo-tagged cargo is retained in the ER by the binding of the Streptavidin binding peptide (SBP) to Streptavidin (StrepA), which is fused to the ER retention signal, KDEL. The cargo is synchronously released upon the addition of biotin, which outcompetes SBP for binding to Streptavidin. **B)** Diagram of the tagged cargo constructs used in this study. A signal sequence is followed by the SBP, then the tag, and the cargo. Cadherin 99C (Cad99c) was used as an apical cargo, and Nidogen (Ndg) as a basal cargo. **C)** Steady state expression of UAS-SBP-Halo-Cadherin99c (magenta) under the control of traffic jam-Gal4, showing its localization to the apical microvilli. Phalloidin staining of F-actin (green) labels the apical microvilli in the follicle cells and the oocyte. Scale bar 10 μm. **D)** Steady state expression of UAS-SBP-SNAP-Ndg (magenta) under the control of traffic jam-Gal4, showing its localization to the basement membrane. Actin is shown in green. Scale bar 10 μm. **E)** En face view of SBP-SNAP-Ndg in the basement membrane. Scale bar 10 μm. **F)** Time course of SBP-Halo-Cadherin99C trafficking in fixed samples. At 0 min, Cad99c localizes to the ER throughout the cytoplasm. Fifteen min after release from the ER with biotin, Cad99c (green) has accumulated in the Golgi, marked by Golgin245 (magenta). By 30 min, almost all Cad99c has reached the apical membrane. Scale bar 10 μm. **G)** 25 min after release from the ER, Cad99c (green) localizes to subapical puncta that are labeled by Rab11 (magenta). Scale bar 10 μm. In all figures with a cross-section of the follicle cells, apical is toward the top of the image and basal toward the bottom.

human Usher cadherin PCDH15 and is expressed in the follicle cells throughout oogenesis, where it localizes to the apical microvilli [16,17]. For the basolateral cargo, we chose SNAP-tag labeled Nidogen, as it is a monomeric extracellular matrix protein that localizes to the basement membrane beneath the follicle cells and is presumably secreted from the bottom third of the lateral membrane like Collagen IV [18–21].

When SBP-Halo-Cad99C is expressed in the follicle cells under the control of traffic jam-Gal4 in the absence of the Streptavidin-KDEL hook and labeled with cell-permeable fluorescent HaloTag ligand, it localizes to the apical domain (Fig 1C)[17,22]. Similarly, SBP-SNAP-Nidogen localizes to the basement membrane like the endogenous protein, demonstrating that both constructs contain functional trafficking signals (Fig 1D and 1E) [18]. Expressing two copies of UAS-Streptavidin-KDEL is sufficient to retain all SBP-Halo-Cad99C in the ER without activating the ER stress response, as monitored by the levels of BiP, a molecular chaperone that is up-regulated in response to ER stress[23] (S1 Fig). On the addition of biotin, it is released and most Cad99c is localized in the

Golgi after 15 min (Fig 1F), partially co-localizing with the *trans*-Golgi marker, Golgin 245 [24], and by 30 min, almost all labeled protein is localized to the apical microvilli, indicating that it rapidly transits the secretory pathway. Cad99c co-localizes with Rab11 in puncta immediately below the apical membrane before it reaches the apical surface, indicating that it passes through the Rab11-positive recycling endosomes en route to the cell surface (Fig 1G)

To characterize the time course of Cad99c and Nidogen secretion, we made time-lapse movies of each labeled cargo after their release from the ER with biotin (Fig 2A and 2B and S1 and S2 Movies). As seen in the fixed samples, most Cad99c localized to the Golgi after 10 min and to apical recycling endosomes after 20 min. Cad99c was detected on the apical surface before 20 min and reached maximal levels after 30 min. At later time points, Cad99c often spread into the lateral membrane (S2 Fig), probably because the over-expressed protein saturated its apical binding sites and diffused in the plasma membrane after secretion. The trafficking of Nidogen to the basal surface showed very similar dynamics to Cad99c. We quantified the dynamics of trafficking through the secretory pathway in more detail by marking ER exit sites, the *cis*-Golgi, the *trans*-Golgi and TGN, and TGN and recycling endosomes with endogenously-tagged Sec16, GMAP-210, Golgin 245 and Rab11 respectively [25–27]. We labeled the plasma membrane with CellMask and measured the change in co-localization of Cad99c with each marker over time using the normalized Pearson correlation coefficient (Figs 2C, 2D, 2G, S2B, S1C and S2D and S3 Movie and S1 Data). Peak levels of Cad99C colocalized with ER exit sites at 11 min and with the *trans*-Golgi after 22–23 min, indicating that it takes about 11–12 min to exit the ER and traverse the Golgi cisternae. Cad99c can be seen colocalizing with Rab11, with peaks at 20 and 25 min as the Cad99c goes from the Golgi to Rab11 recycling positive endosomes. Cad99c first appeared at the plasma membrane after 11 min and the amount of apical protein increased approximately linearly over the subsequent 20 min. Cad99c accumulates at a similar rate at the TGN and at the plasma membrane with a time lag of 6–8 min, indicating that this is how long it takes to traffic protein from the TGN to the cell surface. Cad99c reaches the plasma membrane at a similar rate to that of Serpentine in the fly tracheal epithelium but is significantly faster than cargoes that have been tracked in tissue culture cells using RUSH or VSVG-ts [6,14,28]. This may be because the follicle cells and trachea are intact secretory epithelia rather than transformed tissue culture cells and are therefore more efficient at polarized exocytosis.

We performed a similar analysis on the dynamics of Nidogen secretion, except that we used wheat germ agglutinin as a plasma membrane marker instead of Cell Mask, which does not efficiently label the basal plasma membrane (Figs 2E, 2F, 2H, S2E and S2F and S4 Movie and S1 Data). This revealed that a significant proportion of Nidogen already localizes to ER exit sites before release, which may reflect the fact that Nidogen is a soluble cargo and can therefore concentrate in the ER lumen near the exit sites. Nidogen also transits the Golgi in < 4 min, which is much faster than Cad99c (~10 min). Soluble cargoes have been observed to transit the Golgi more quickly than membrane-spanning cargoes in a variety of contexts, but the reason for this difference is still a subject of debate [29–32]. Perhaps as a consequence, Nidogen starts to accumulate extracellularly slightly earlier than Cad99c. Large ECM cargoes like Collagen IV have been found to leave the ER through specialized ER exit sites (ERES) [33]. We therefore tested whether this is also the case for Nidogen by co-expressing SBP-SNAP-Nidogen and SBP-HALO-Cad99c and determining whether Nidogen leaves the ER through distinct ERES from Cad99c. However, we observed that all ERES that are positive for Cad99c were also labeled by Nidogen, indicating that a soluble ECM cargo and an apical transmembrane protein exit the ER through the same sites (S3 Fig).

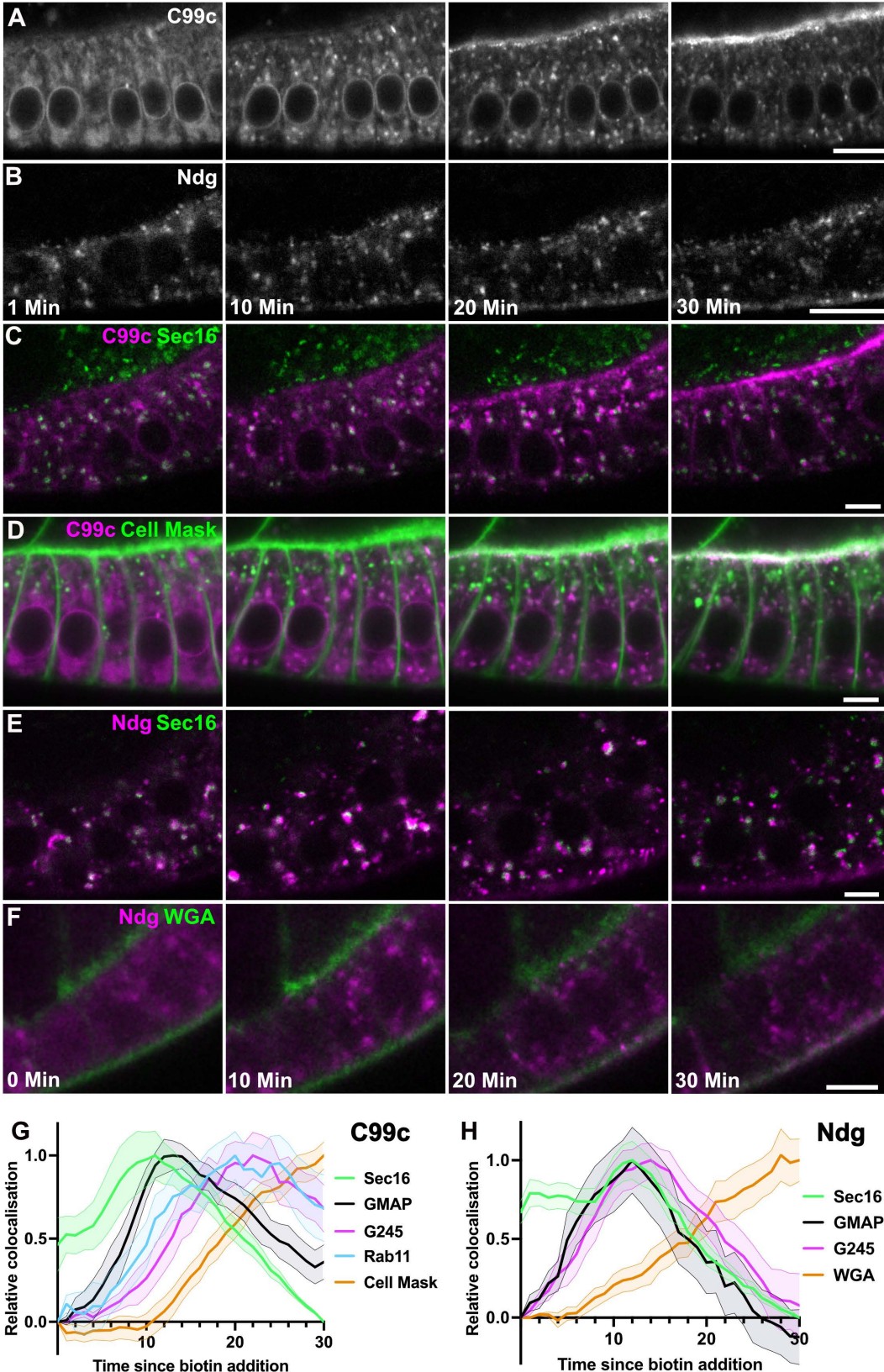

**Fig 2. Time course of Cad99c and Nidogen secretion. A-B)** Stills from RUSH movies of SBP-Halo-Cadherin 99c (A) and SBP-SNAP-Nidogen (B) trafficking in the follicle cells after release from the ER. Cad99c is shown in an early stage 9

egg chamber, and the Ndg in a stage 8 egg chamber. Scale bars 10 μm. Time indicates minutes since the addition of biotin. Cad99c is labeled with Halo-OregonGreen, and Ndg with SNAP-SiR. The large cytoplasmic puncta visible 10 min after cargo release are Golgi ministacks. **C)** Stills from a RUSH movie of SBP-Halo-Cadherin 99c (labeled with Halo-JF646; magenta) in cells expressing GFP-Sec16 (green), which marks ER exit sites. Scale bars 5 μm. Cad99c partially co-localizes with Sec16 at ER exit sites after 10 min, and by 20 min is predominantly localized to the Golgi, which lie adjacent to the ER exit sites (see S2 Fig). **D)** Stills from a RUSH movie of SBP-Halo-Cadherin 99c (magenta) in cells in which the plasma membrane is labeled with CellMask. Some Cad99c has already reached the apical membrane by 20 min and much more is localized apically at 30 min. **E)** Stills from a movie of SBP-Halo-Ndg RUSH in cells expressing GFP-Sec16. Some colocalization of Ndg with Sec16 can be seen at time 0, presumably as Ndg is a soluble protein present in the ER, and is able to partly accumulate at the ER exit sites. Most Nidogen localizes at ER exit sites after 10 minutes and in the adjacent Golgi mini-stacks after 20 min. **F)** Stills from a movie of SBP-Halo-Ndg RUSH (SNAP-SiR; magenta) in cells labeled with alexa488-wheatgerm agglutinin (WGA; green), which labels the basal membrane. Scale bars 5 μm. Colocalization of Ndg with WGA can be seen from 20 min, and is stronger at 30 min as Ndg has been secreted and bound to the ECM. **G)** Graph showing the change in Cad99c colocalization with GFP-Sec16 (ER exit sites), GFP-GMAP (*cis*-Golgi), GFP-G245 (*trans*-Golgi and TGN) and CellMask (plasma membrane) over time. Peak colocalization has been normalized to 1 to allow easier comparison between markers. Cad99c colocalization with Sec16 peaks at 11 min after biotin addition, its colocalization with GMAP peaks between 12–14 min and with G245 is at 22–23 min. Cad99c starts to co-localize with CellMask at 11 min and increases over time. The number of movies used for each marker is detailed in Table 2. Shaded areas indicate ± the s.e.m. **H)** Graph showing the change in Nidogen colocalization with GFP-Sec16, GFP-GMAP, GFP-Golgin245, and with WGA, which marks the basal membrane. Peak colocalization has been normalized to 1 to allow easier comparison between markers. Ndg colocalization with Sec16 is already high before biotin addition, and peaks at 12 min after biotin addition, which is also when the peak colocalization with GMAP occurs, indicating rapid transit of Ndg from the ER exit sites to the *cis*-Golgi. Peak Ndg colocalization with G245 is at 14 min, again indicating rapid transit through the Golgi. Basal Ndg signal is observable from 6 min, and increases over time. It is important to note that as a soluble protein, the Ndg present in the basement membrane is not necessarily from the egg chamber being observed. The number of movies used for each marker is detailed in Table 2. Shaded areas indicate ± the s.e.m. Data for G) and H) can be found in S1 Data.

## The MSP-tracker and MSP-viewer programs

To examine the movement of post-Golgi carriers from the TGN to the plasma membrane, we performed fast time-lapse imaging at 2–4.5 frames/second of labeled cargoes 15–20 min after the addition of biotin, i.e., just after their peak accumulation in the TGN. This revealed that many Cad99c positive vesicles show processive movements towards the apical surface, whereas Nidogen-containing vesicles show more randomly-directed trajectories. However, tracking these movements was laborious and challenging due to the large number of small vesicles, their low signal to noise ratio compared to the much brighter Golgi and their dispersed origins from Golgi mini-stacks throughout the cytoplasm (Fig 3A). Many different tracking algorithms have been developed in recent years that use deep learning (DL) methods to enhance target detection and object tracking [34–47]. However, these machine learning methods usually require an extensive training dataset that must be updated for each new experimental system, which can be impractical and time consuming, some were developed for particular imaging techniques, while others perform poorly in a noisy environment. DL methods have also been integrated into several tracking platforms [48], including Track-Mate [43] DeepTree [49], ELEPHANT [50] and others, but these are mainly designed for cell segmentation, rather than tracking small fast-moving particles. To address these drawbacks, we have developed an multiscale particle (MSP) framework to facilitate particle tracking in a noisy environment.

MSP-tracker is an extended version of our particle tracking software that uses a convolutional neural network (CNN) for candidate vesicle selection and a two-step process to identify tracklets and then link them into trajectories [51]. The particle detection module consists of background subtraction step followed by the identification of spots of different sizes and brightness using a multiscale spot enhancing filter (MSSEF) [52] (Fig 3B). This first step creates a pool of candidate vesicles, but is rather noisy, as MSSEF cannot distinguish between post-Golgi vesicles and Golgi ministacks or other background signals. A light-weight CNN classifier then selects vesicles

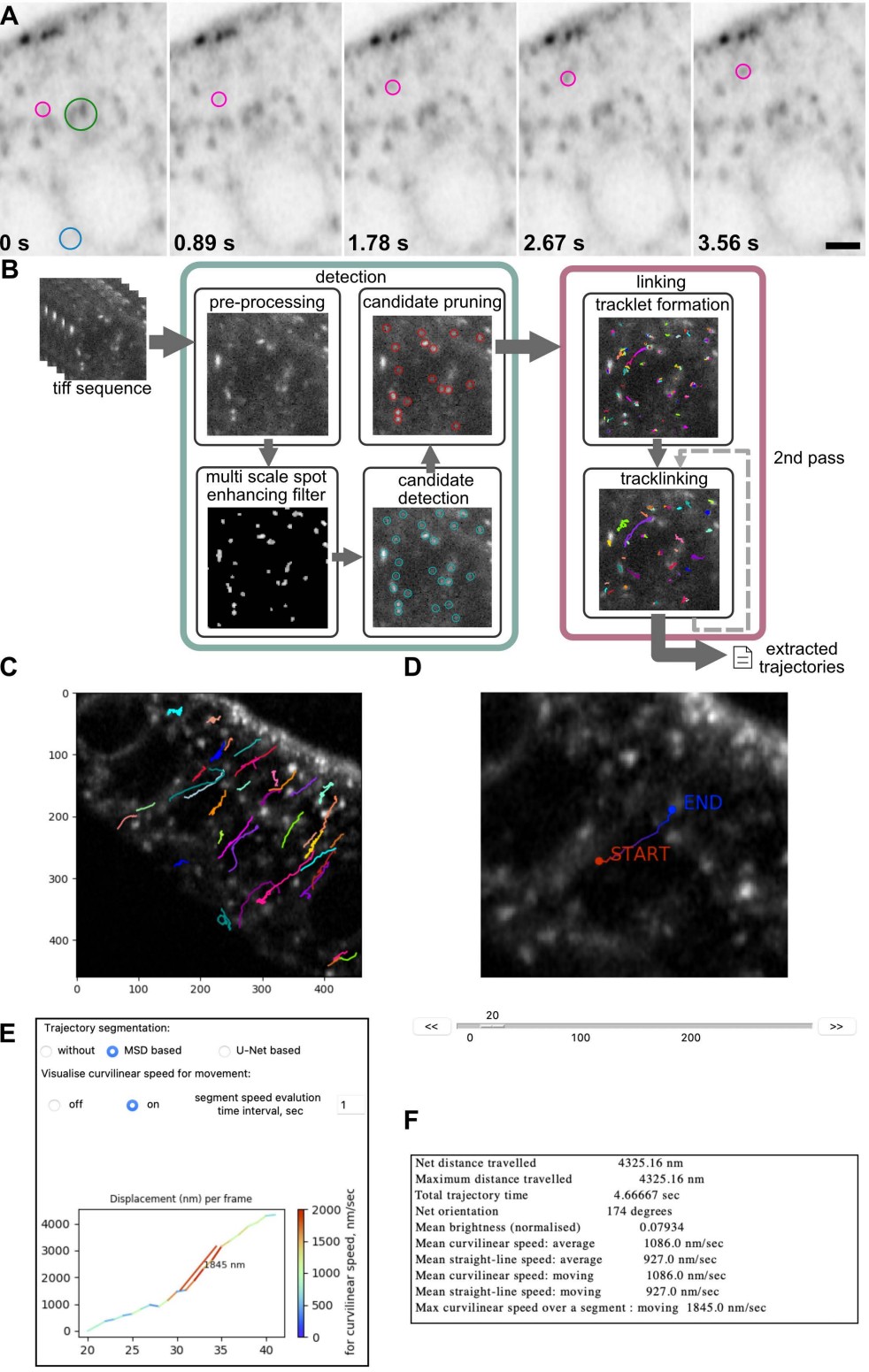

**Fig 3. The MSP-tracker and viewer analysis pipeline. A)** Examples of frames from a Cad99c RUSH movie, highlighting both particles of interest (magenta circles), background/ER signal (blue circle), and Golgi mini-stacks (green circle). Scale bar 2 μm. **B)** Workflow for MSP-Tracker showing the steps taken from .tiff file to trajectories. **C)** An example of the output of MSP-viewer showing the trajectories of Cad99c vesicles overlaid onto a movie still. The image was obtained using the trajectories: save image command in the mspviewer software. **D)** Example of a Cad99c

track as seen in the "trackviewer" window of MSP-viewer. **E)** Example of the trajectory segmentation section of the "trackviewer" window in MSP-viewer. The marked speed is the fastest segment of the track over a 1 second time interval. **F)** Trajectory information as viewed in the "trackviewer" window. These data are output into a.csv file using the trajectories: save info to csv function.

from the pool. The decreased number of parameters in the CNN compared to other approaches reduces the number of annotations required for training to a few hundred regions of interest (ROIs) and provides a fast candidate pruning process that significantly improves detection efficiency. The MSP-framework comes with a few networks that have already been trained, but the CNN classifier can be easily trained on new data if required. In the last step, the locations of the selected candidates are refined to provide subpixel localization. Due to its computational efficiency, this step uses the iterative center of-mass refinement [42]. This pruned and refined set of detected particles is then handed to the linking algorithms.

The detected particles in different frames are joined into tracks with a two-step linking approach. First, short tracks (tracklets) are formed based on the vesicles' proximity in time and space. These short tracklets are then linked based on their characteristics, in a process called track-linking. Given the complex and changeable behavior of the particles, this multistep approach builds trajectories without assumptions about the type of movement. The first step takes into account the distance between detections and existing tracks, solving the assignment problem sequentially for each frame. The problem can be described as minimum-weight matching and is solved with the Hungarian algorithm [53]. Detections that are not matched with any of the tracks start their own new track. The length of the tracklet is limited, as well as the number of the sequentially missed frames. This way, short, reliable tracklets are formed based on the spatial and temporal distances.

A Bayesian network (BN) then calculates a connectivity score for each pair of tracklets, based on their positions in time and space, speeds and directions of movement and their appearances. The connectivity score is only calculated for tracklets that are close in space and time, as this significantly reduces computation time and increases the efficiency of the method. When all the possible tracklet pairs have been evaluated, the tracklets are assembled into tracks using the Hungarian algorithm [53]. The minimum number of points in the trajectory can be set manually by the user and it can be useful to filter by length to eliminate very short trajectories, which can arise from false detections. MSP-tracker allows users to repeat the track-linking process with different settings to link tracks into larger trajectories, as a second pass can be beneficial in a dense environment or when some particles move much faster than the majority. In these cases, the first pass connects slow-moving particles (using a small value for the distance parameter) and the second pass connects faster-moving ones (using a larger value for the distance parameter). The software displays the intermediate results from the vesicle detection and track-linking algorithms, revealing how parameter selection affects the output. The optimal parameters can then be saved in an external file for future reference. Finally, the extracted tracks are stored in a format compatible with MSP-viewer, where the trajectories can be examined (Fig 3C).

MSP-viewer is the tool for exploring extracted tracks and allows the user to view the tracks, correct them if required, plot them and analyze their characteristics. MSP-viewer is also a convenient tool for exploring tracksets exported from other sources. The main window displays the image sequence overlaid with the plotted trajectories and a table of the individual trajectories (Fig 3C). The set of trajectories can then be filtered based on their duration (length), net or total distances travelled, mean curvilinear speed, or trajectory orientation. The results can also be displayed as a circular plot of trajectory orientations based on track count or net distance travelled. Clicking on an individual track in the table opens a new window that shows the trajectory, a graph of the speed of vesicle movement over time and a table listing the net distance travelled, the mean curvilinear and straight line speeds over the whole trajectory

and when the particle is moving, as well as the maximum curvilinear speed (Fig 3D–3F). The code for both MSP-tracker and MSP-viewer, video tutorials, and a detailed manual with step-by-step instructions for installation and parameter settings are available at https://doi. org/10.17863/CAM.114339).

## Evaluation of MSP-tracker

To evaluate the performance of MSP-tracker under controlled conditions, we used 12 image sequences of 100 frames of computer-generated data from the particle tracking challenge [37]. The scenario mimics real data of moving vesicles at three density levels (low, middle, and high), and at four different signal-to-noise ratios (SNR) of Poisson noise (1, 2, 4, and 7) (Fig 4A). We compared the performance of MSP-tracker to the tracking methods that participated in the particle tracking challenge using the five performance measures used in the original challenge paper (Fig 4B) [37]. Linking performance is quantified by the degree of matching between the ground truth and estimated tracks, expressed by the $\alpha$ and $\beta$ measures, and by two Jaccard similarity coefficients, one for entire tracks, $JSC_{\theta}$, and one for track points, JSC. The $\alpha$ measure does not take into account spurious tracks and ranges from 0 (worst) to 1 (best), while the $\beta$ measure has a penalty for spurious tracks, and therefore ranges from 0 to $\alpha$. Both Jaccard similarity coefficients range from 0 to 1. The particle localization accuracy is evaluated by the root mean square error (RMSE) of matching points in the paired tracks. The lines in Fig 4A represent the values for MSP-tracker and the colored areas show the range of scores reported in the original challenge paper [37].

The performance of MSP-tracker falls within the upper range of the published results of the particle tracking challenge for the $\alpha$ and $\beta$ measures but gives lower results for high-density samples in the Jaccard similarity coefficient for track points, JSC. This may reflect the fact that the CNN was not trained on this type of artificial data and was developed to detect the vesicles of interest against a background of other fluorescent objects in the cell, such as the Golgi. However, MSP-tracker performs well for the Jaccard similarity coefficient for the entire tracks, $JSC_{\theta}$. This highlights its ability to link detections into the correct tracks, even if the detections inside the tracks are not complete.

A more relevant test for MSP-tracker is to evaluate its performance on real data that has been manually tracked to determine the ground truth. We therefore analyzed whether MSP-tracker correctly identified tracks in 12 video sequences containing a total of 391 annotated tracks. To examine the performance of the tracker under different conditions, we included movies taken using both Airyscan and spinning disc microscopes, different RUSH cargoes, and imaged along cross-sectional and planar views of the egg chamber. The ground truth data was manually annotated using the ImageJ multipoint tool [54] and MSP-viewer, which provides a faster and more convenient platform for tracing vesicle trajectories. The manually tracked data do not provide subpixel particle localizations unlike automated methods, and we therefore did not use the RMSE measure for the evaluation.

The results from MSP-tracker were compared with the performances of the most recent version of Trackmate [40,43], which is widely used as an ImageJ plugin and SpotTracking [55], which scored in the top 3 in most categories of the particle tracking challenge[37]. In addition to the linking measures that have already been mentioned, $\alpha$, $\beta$, JSC, and $JSC_{\theta}$, we used a novel measure $\gamma$, which represents the proportion of manually annotated tracks that were detected by the software. Parameters were set individually for each video sequence to achieve the best performance in each software. MSP-tracker produced significantly better results than the two other tracking programs in all metrics (Fig 4C). This is mainly due to the detection component of the tracker, where the candidate pruning improves detection and therefore boosts tracker performance. Nevertheless, all three programs detect some spurious

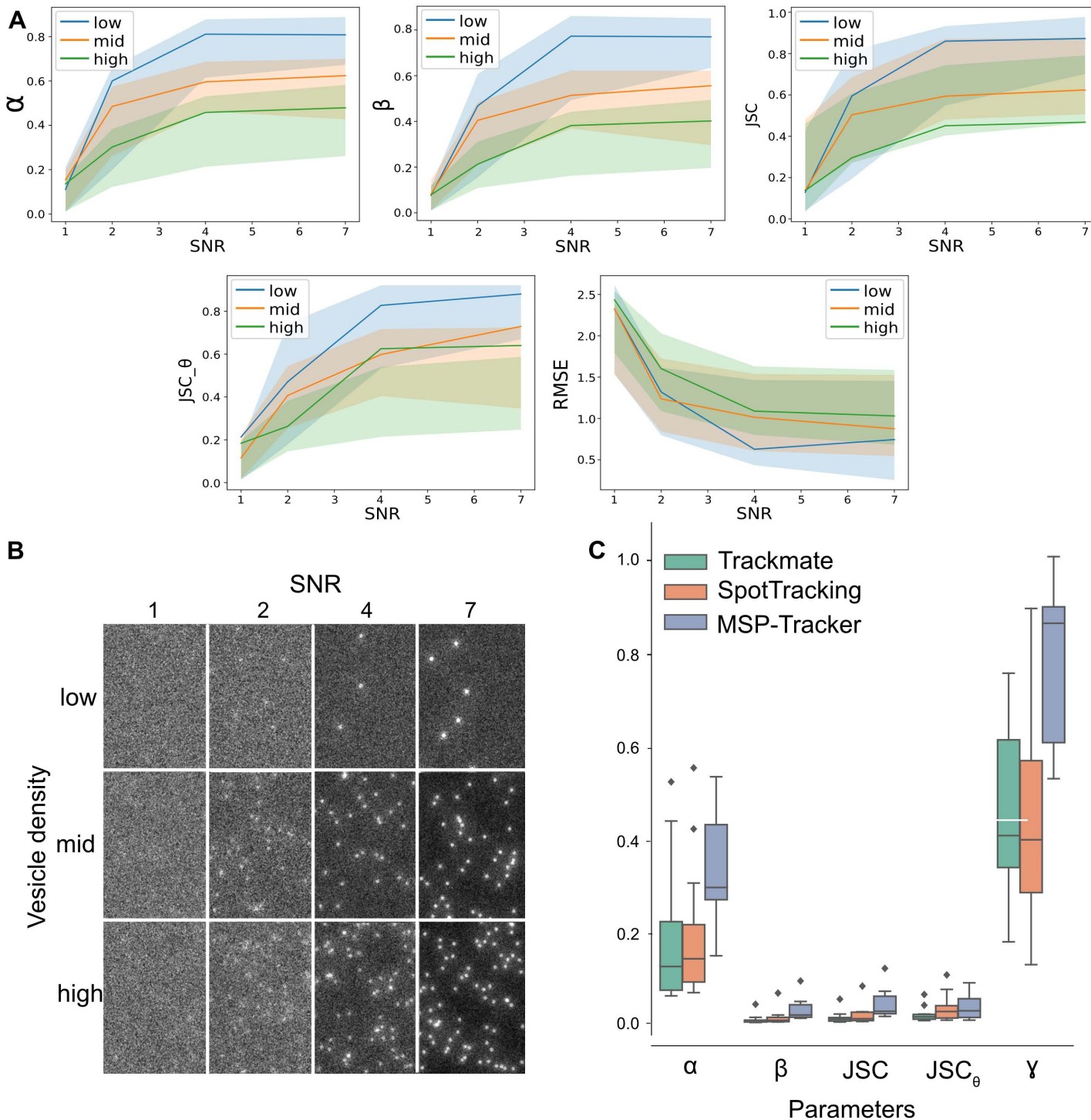

**Fig 4. Evaluation of the MSP-tracker. A)** Performance of MSP-tracker evaluated on the particle tracking challenge data [37]. The performance is evaluated by α and β measures, the Jaccard similarity coefficient for track points JSC, the Jaccard similarity coefficient for the entire tracks $JSC_\theta$ and root mean square error (RMSE) of matching points in the paired tracks. The lines represent the MSP-tracker results and colored areas refer to the range between the highest and lowest scores in the original challenge paper [37]. **B)** Frame examples of the particle tracking challenge data. **C)** Performance of the proposed MSP-tracker, Trackmate and SpotTracking evaluated on the original data. Linking performance is evaluated by α and β measures, the Jaccard similarity coefficient for track points JSC, as well as for the entire tracks $JSC_\theta$ and a fraction of paired up tracks γ. Data for A) and C) can be found in S2 Data, and at https://doi.org/10.17863/CAM.114338.2.

tracks, which is reflected in the low $\beta$ and $JSC_\theta$ scores. This is largely due to the nature of the data itself. The dispersed mini-Golgi stacks can resemble vesicles, while some vesicles, located away from the focal plane appear as blurry and noisy areas that are challenging to detect. Given the uncertainty about how many of these "false positive" tracks are genuine, $\gamma$ represents a more meaningful performance metric, since it measures the proportion of verified tracks that each software detected. MSP-tracker performed much better than the other programs on this metric, as it detected nearly 90% of the annotated tracks compared to 40% for Trackmate and SpotTracking (Fig 4C and S2 Data). The dataset and parameters used are available https://doi.org/10.17863/CAM.114338.2

A key part of our analysis pipeline is to use MSP-viewer to manually curate the tracks after the automated track detection by MSP-tracker. We also manually add in any tracks that have been missed by MSP-tracker, but are visible to the human eye. MSP-viewer provides a straightforward and efficient platform to do this, resulting in a final track list that is as close to the ground truth as possible.

## Tracking Cadherin 99c and Nidogen vesicles

We used automated tracking with manual curation to analyze the direction of Cad99c-positive vesicle movements at different stages of oogenesis and at different time points after the addition of biotin. At stage 8 when the follicle cells are still cuboidal, Cad99c-positive vesicles move in all directions. To quantify this, we counted the number of apically-directed tracks (defined as tracks whose orientation fell within 45° of apical), compared to lateral (left and right 90° quadrants) and basal. 31% of vesicles moved apically, 52% moved laterally and 17% basally (Fig 5A, S1 Table, S5 Movie). The apical tracks are on average longer than the basal ones, however, and this results in a mean apical displacement per track of 0.49 μm ± 0.26 (s.e.m), which is sufficient to deliver Cad99c to the apical plasma membrane (Fig 5D). The columnar follicle cells at stage 9 show a much stronger directional bias towards the apical surface, with 73% of the Cad99c-positive vesicles moving apically compared to 17% basally, giving a mean apical displacement per track of 1.32μm ± 0.18 (s.e.m) (Fig 5B and 5D, S1 Table, S5 Movie, S3 Data). This increase in directionality correlates with the increase in the abundance of long microtubules that extend along the apical-basal axis of the columnar follicle cells compared to the cuboidal cells at stage 8 (S7A, S7C, S7E and S7G Fig) [22,56,57]. The strong directional bias in the movement of Cad99c vesicles starts to diminish after 30 min post biotin addition, when most Cad99c has already reached the apical membrane, presumably because secretion is now balanced by endocytosis in the opposite direction (Fig 5C, S5 Movie), indicating that MSP-tracker can also map the trajectories of endocytic vesicles. We saw the same trend when tracking vesicle movements imaged on a spinning disc microscope, with a strong apical bias in Cad99c tracks (S4A Fig and S4 Data) that is lost when imaging later after biotin addition (S4B Fig). A similar analysis of the trajectories of Nidogen vesicles revealed that they move in more random directions with a slight bias towards the basal side (Figs 6A and S5A, S1 Table, S6 Movie, S3 and S4 Data files). 33% of vesicles move towards the basal quadrant, compared to 25% apically and 41% laterally (S1 Table). Nevertheless, the weak directional bias leads to a mean basal displacement per track of 0.17μm ± 0.09 (s.e.m) (Fig 6B).

Studies on the secretion of another ECM protein, Collagen IV, in the follicle cells indicate that it is secreted in a planar-polarized manner from the basal third of the lateral membrane and that it is transported there in Rab10[+ve] vesicles by kinesin-1 and Khc73 [20,21]. During stages 3–8 of oogenesis, the follicle cells collectively migrate over the underlying basement membrane, causing the egg chamber to rotate around the anterior-posterior axis, in a process that aligns the collagen fibrils so that they form a molecular corset around the dorsal ventral axis [21,58]. Imaging of Rab10[+ve] vesicles revealed that these move towards the trailing edges

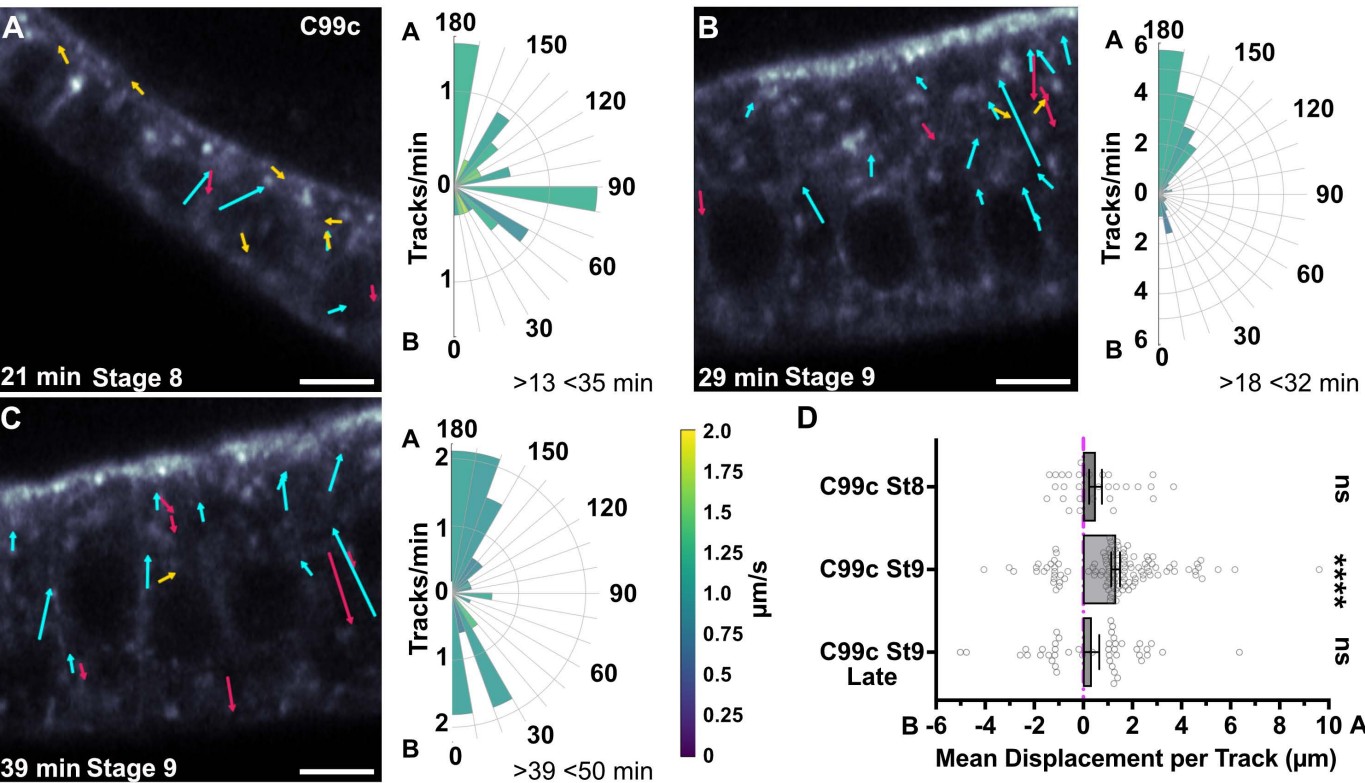

**Fig 5. Cad99c tracks are apically biased.** Arrows superimposed on movie stills indicate the vectors of the vesicle tracks from the movie. Cyan arrows are apically directed, red arrows are basally directed, and yellow arrows are laterally directed. **A)** Tracks from a representative movie of Cad99c trafficking in a stage 8 egg chamber 21 min after biotin addition. The polar histogram on the right shows the orientation of Cad99c tracks from 3 Airyscan movies. This and the following polar histograms represent the number of tracks that traveled a net distance of at least 1 μm in each 10° direction, with apical (A) at the top and basal (B) at the bottom. The plots are normalized to show tracks per minute and speeds are color coded according to the key. All movies were collected at 4.5 fps. Cad99c was labeled with Halo-JF549. Scale bars are 5 μm. **B)** Tracks from representative movie of Cad99c trafficking in a stage 9 egg chamber, 29 min after biotin addition. The polar plot represents data from 5 Airyscan movies and shows a strong apical bias in the direction of Cad99c vesicle movements. **C)** Tracks from representative movie of Cad99c trafficking in a stage 9 egg chamber 39 min after biotin addition. The polar plot represents data from 3 airyscan movies. By this time point, the strong apical bias is lost, as the exocytosis of C99c is balanced by endocytosis. **D)** Graph showing the net displacement of Cad99c vesicle tracks along the apical-basal axis, with positive displacement towards apical (A) and negative towards basal (B). The bar shows the mean, and the error bars the s.e.m. Individual track displacements are indicated by the small circles. The mean displacement for stage 8 is 0.49 μm ± 0.26 (s.e.m.), and for stage 9 is 1.32 μm ± 0.18 (s.e.m.). Significance was calculated using a Wilcoxon Signed-Rank test for comparing the displacement to zero. Track data can be found in S3 Data. Movies used for tracking can be found at https://doi.org/10.17863/CAM.114338.2.

of the migrating cells, where Collagen IV is thought to be secreted [20,59]. We therefore examined whether Nidogen carriers also show planar-polarized movements basally by imaging horizontal planes of the epithelium either in the middle of the cell or just above the basal surface (Fig 6C and 6D, S6 Movie, S3 Data). The vesicle tracks showed little planar-polarized bias in their directions of movement in the middle of the cell, but showed a pronounced planar polarization basally, with 45% moving towards the leading edge, compared to 30% towards the trailing edge (S1 Table). Surprisingly, this is the opposite bias to that reported for Rab10[+ve] vesicles, which are presumed to contain Viking (Collagen IV) [20]. Movies collected on the spinning disc microscope showed the same bias (S5 Fig).

To test whether Nidogen is transported to the leading edge in Rab10[+ve] vesicles, we co-expressed UAS-Rab10-YFP with the Nidogen RUSH construct. Rab10 co-localizes with Nidogen in large vesicular compartments at the base of the cells and remains associated with Nidogen vesicles after they leave this structure (Fig 7A and 7B, S7 Movie, S3 Data). Fast imaging of the basal side of the cells 15 min after the release of Nidogen with biotin revealed that

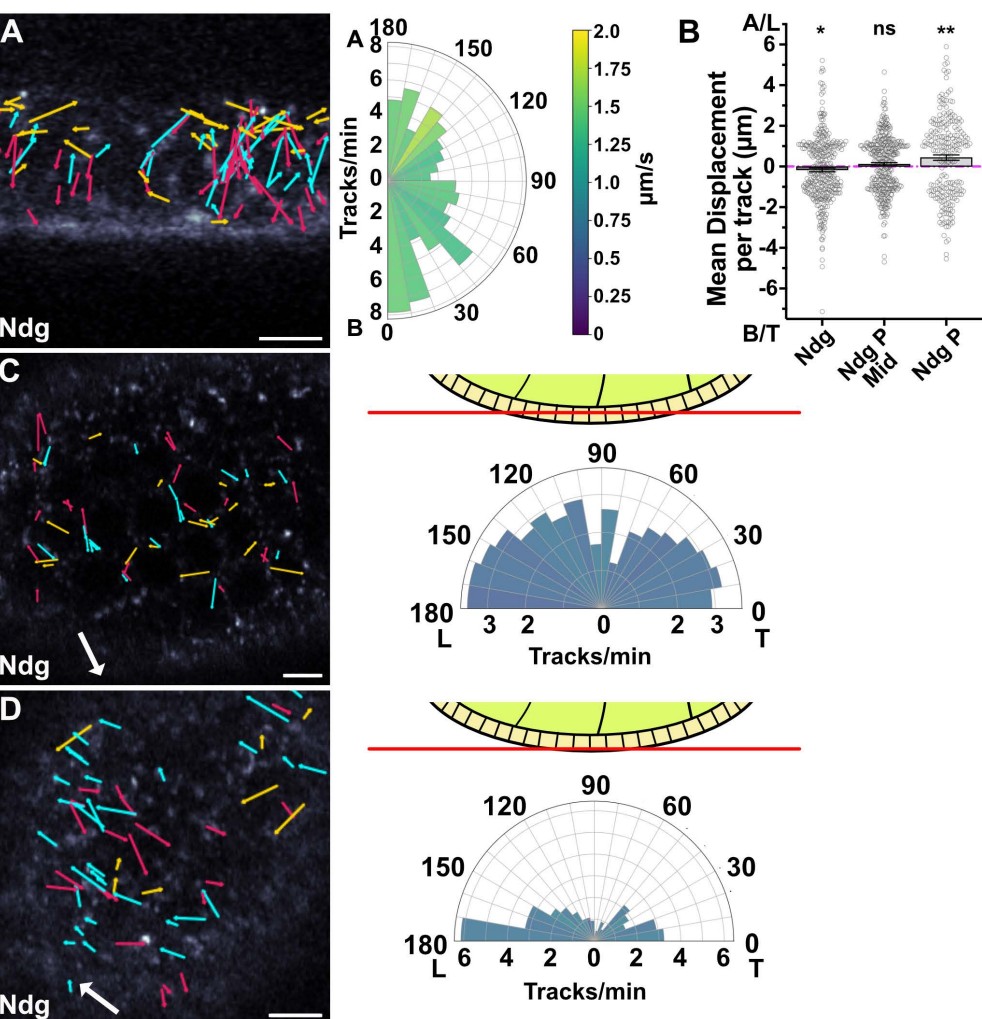

**Fig 6. Analysis of Ndg tracks reveals planar polarized transport basally. A)** Tracks from a representative movie of Ndg trafficking in a stage 8 egg chamber 23 min after biotin addition. The polar plot on the right represents data from 3 Airyscan movies and shows a weak bias towards the basal side of the cell. Speeds are color-coded according to the key. Movies were taken at 3.7 fps. Arrows superimposed on movie stills indicate the vectors of the vesicle tracks from the movie. Cyan arrows are apically directed, red arrows basally directed and yellow arrows laterally directed. **B)** Graph showing the net displacement of Ndg vesicle tracks along the apical-basal axis (A, B; lefthand column) and along the leading edge to trailing edge axis (L,T) in the middle of the cell (middle column) and on the basal side of the cell (righthand column). The bars and error bars show the means and s.e.m. Individual tracks are indicated by the small circles. P indicates planar. Mean displacement towards basal is 0.17 μm ±0.09 (s.e.m). The mean displacement for planar polarized tracks in the middle of the cell was 0.11 μm ±0.07 (s.e.m) towards the leading edge, and for the basal tracks, 0.44 μm ±0.13 (s.e.m) towards the leading edge. Significance was calculated using a Wilcoxon Signed-Rank test comparing the displacement to zero. **C)** Ndg tracks in a transverse section through the middle of the follicle cell epithelium (see diagram). The large white arrow indicates the direction of egg chamber rotation and the coloured arrows indicate the vectors of vesicle tracks, with tracks towards the leading edge in cyan, towards the trailing edge in red and tracks perpendicular to the axis of rotation in yellow. The polar plot on the right represents data from 3 Airyscan movies taken at 2.3 fps. The leading and trailing (L and T) directions of the egg chamber's rotation are indicated. **D)** Ndg tracks in a transverse section through the basal section of the follicle cell epithelium. The direction of rotation and the tracks are labeled as in (C). Scale bars 5 μm. The polar plot represents data from 3 Airyscan movies taken at 2.3 fps. Track data can be found in S3 Data. Movies used for tracking can be found at https://doi.org/10.17863/CAM.114338.2.

many moving Nidogen-containing vesicles were also Rab10[+ve] (Fig 7C; S7 Movie). As Rab10 signal bleached easily, and not all Rab10 tracks survived filtering, we loaded the Nidogen track coordinates onto the Rab10 movies in MSP-viewer and manually determined whether

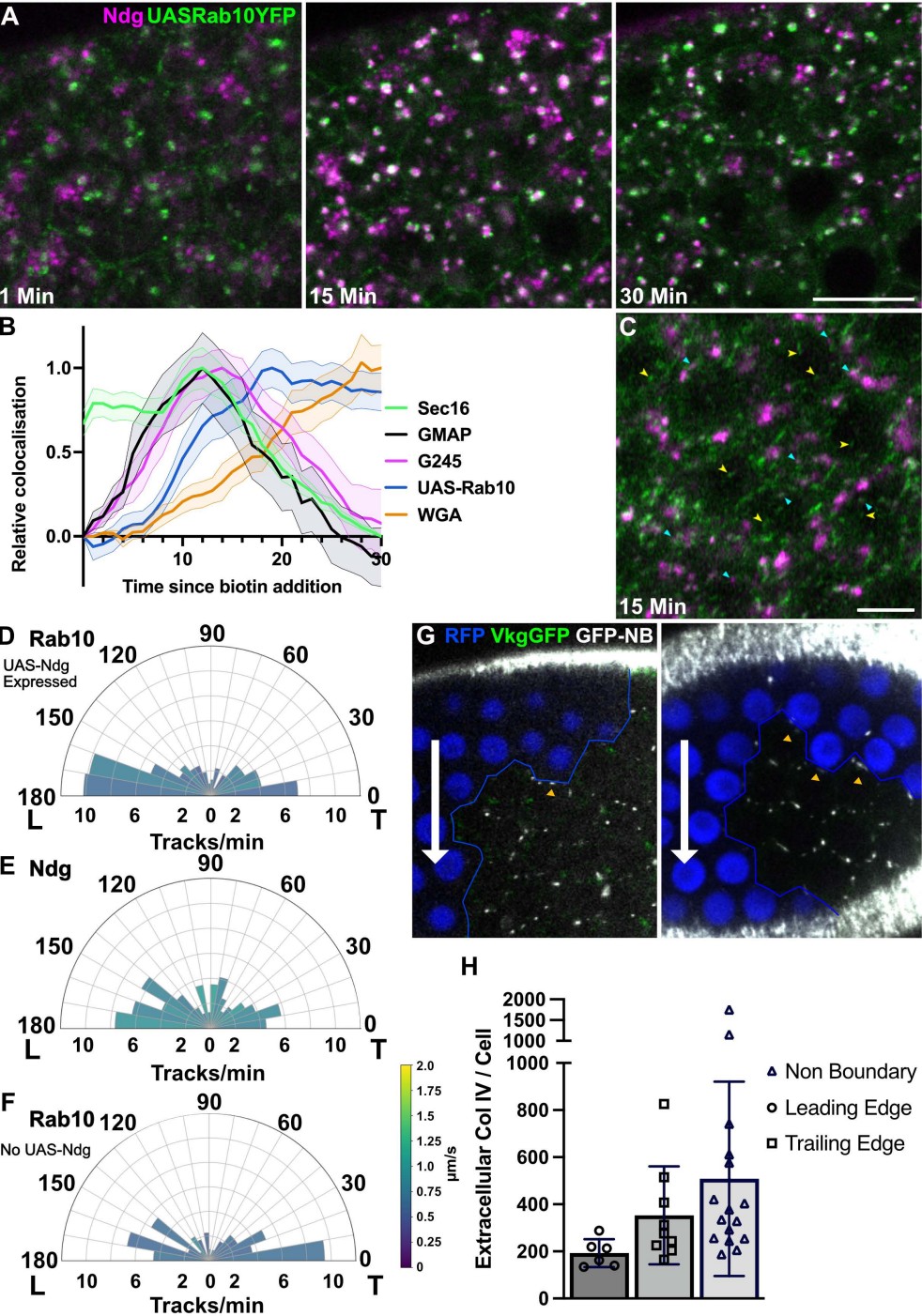

**Fig 7. Ndg is transported in Rab10 positive vesicles. A)** Stills from a RUSH movie of SBP-SNAP-Nidogen (labeled with SNAP-SiR; magenta) in cells expressing UAS-Rab10-YFP (green). Scale bar 5 μm. Significant colocalization of Ndg with Rab10 in large vesicular structures is seen 15 min after biotin addition, but this colocalization has largely disappeared by 30 min. **B)** Graph showing the change in Nidogen colocalization with GFP-Sec16, GFP-GMAP, GFP-Golgin245, UAS-Rab10-YFP and basal WGA over time. Peak colocalization is set to 1 to enable comparison between markers. Ndg colocalization with Rab10 peaks at 19 min, after the peak with G245, indicating Ndg leaves the Golgi and transits through a Rab10 positive compartment on its way out of the cell. The graph is the same as Fig 2H, but with the addition of UAS-Rab10-YFP to show its position in the pathway. Shaded areas indicate ± the s.e.m. **C)** Still from an Airyscan movie of the basal side of cells expressing UAS-Rab10-YFP and Ndg. The yellow arrowheads indicate examples of Rab10 vesicles; cyan arrowheads indicate examples of Rab10 vesicles that contain Ndg. **D)** A

polar plot of Rab10 tracks from movies at the basal side of cells also expressing SNAP-Ndg. Rab10 vesicle movements show a bias towards the leading edge. The polar plot represents tracks from two Airyscan movies. **E)** A polar plot of Ndg tracks from the same movies as D), also showing a leading edge bias. **F)** A polar plot of Rab10 tracks from movies of cells not expressing SNAP-Ndg, showing a bias towards the trailing edge. The plot represents tracks from two Airyscan movies. Speeds are colour coded according to the key. Track data for D), E) and F) can be found in S3 Data. Movies used for tracking can be found at https://doi.org/10.17863/CAM.114338.2. **G)** Stills from movies looking at Collagen IV secretion from Vkg-GFP clones. Double RFP (Vkg-GFP negative) cells that do not express Collagen IV are marked in blue, and the clone boundary is marked by a blue line. The direction of migration in each movie is indicated by a white arrow. Extracellular Collagen IV was labeled with GFP-booster conjugated to Atto647n. Intra- and extracellular Collagen IV is marked with GFP. **H)** Quantification of extracellular Collagen IV levels at the leading and trailing edges of cells, from movies such as those represented in G). Cell whose leading edges lie at the boundary with nonexpressing double RFP cells were used to measure the amount of Collagen IV secreted at the leading edge, cells with trailing edges at the boundary were used to measure the amount of Collagen IV secreted at the trailing edge, and nonboundary cells were used to measure the total amount of secreted Collagen IV. The majority of the Collagen IV is secreted from the trailing edge. Data for H) can be found in S5 Data.

we could see Rab10 signal along the trajectory and moving with the trajectory. Over 75% of Nidogen tracks had Rab10 associated with them for at least part of their length. Tracking Rab10[+ve] vesicles in these movies revealed a similar directional bias to Nidogen vesicles, with 45% moving towards the leading edge (Fig 7D and 7E, S1 Table, S7 Movie). This is the opposite bias to that observed by Zajac and Horne-Badovinac (2022) in cells that did not over-express Nidogen. We therefore tracked Rab10[+ve] vesicles in cells without UAS-Nidogen and observed a bias towards the trailing edge (Fig 7F, S7 Movie). Thus, releasing a pulse of Nidogen through the secretory pathway shifts the direction of Rab10[+ve] vesicle movement towards the leading edge. This confirms that Nidogen is trafficked towards the leading edge of the follicle cells in Rab[10+ve] vesicles.

Since these results above suggest that Nidogen and Collagen are transported to opposite sides of the cell in Rab[10+ve] vesicles, Rab10 cannot be used as a robust marker for Collagen IV trafficking. We therefore set out to confirm that Collagen IV is secreted from the trailing edges of the cells. Since Collagen IV is a trimer, it was not possible to use RUSH to examine its movement within the cell. However, it accumulates laterally between cells, before being drawn into alignment in the basal ECM by attachment to the existing ECM and follicle cell rotation [21]. We therefore imaged egg chambers containing clones of cells lacking GFP-Collagen IV surrounded by cells expressing GFP-Collagen IV from the *viking* protein trap line. By labelling the extracellular Col IV in living egg chambers using a fluorescently-labeled anti-GFP nanobody, we could examine from which side of the cell Col IV is secreted while observing the direction of rotation. More lateral GFP-Col IV could be detected at trailing edges that faced GFP-Col IV nonexpressing cells than at leading edges (Fig 7G and 7H, S5 Data). Thus, follicle cells transport Nidogen and Col IV in opposite directions along the basal array of microtubules to bias their secretion at the leading and trailing edges, respectively.

### The role of microtubules in Cadherin 99c and Nidogen transport

We examined the effects of Cad99c trafficking on disrupting microtubule-based transport. In most epithelial cells, microtubules are nucleated from noncentrosomal microtubule organizing centers at the apical cortex and form apical-basal arrays with their minus ends apical [57,60–62]. Cad99c carriers are thought to be transported along these microtubules by the minus end directed motor protein, Dynein [22]. We therefore used the MSP package to analyse Cad99c carrier movements in stage 8 and 9 follicle cells heterozygous for a dominant EMS-induced mutation in the Dynein heavy chain, $Dhc^{8-1}$, which reduces the speed of the dynein-dependent transport of *gurken* and *bicoid* RNAs by 50% and 36%, respectively [63–65] (Figs 8A and S6A, S8 Movie, S3 Data). We confirmed that this mutation does not affect the

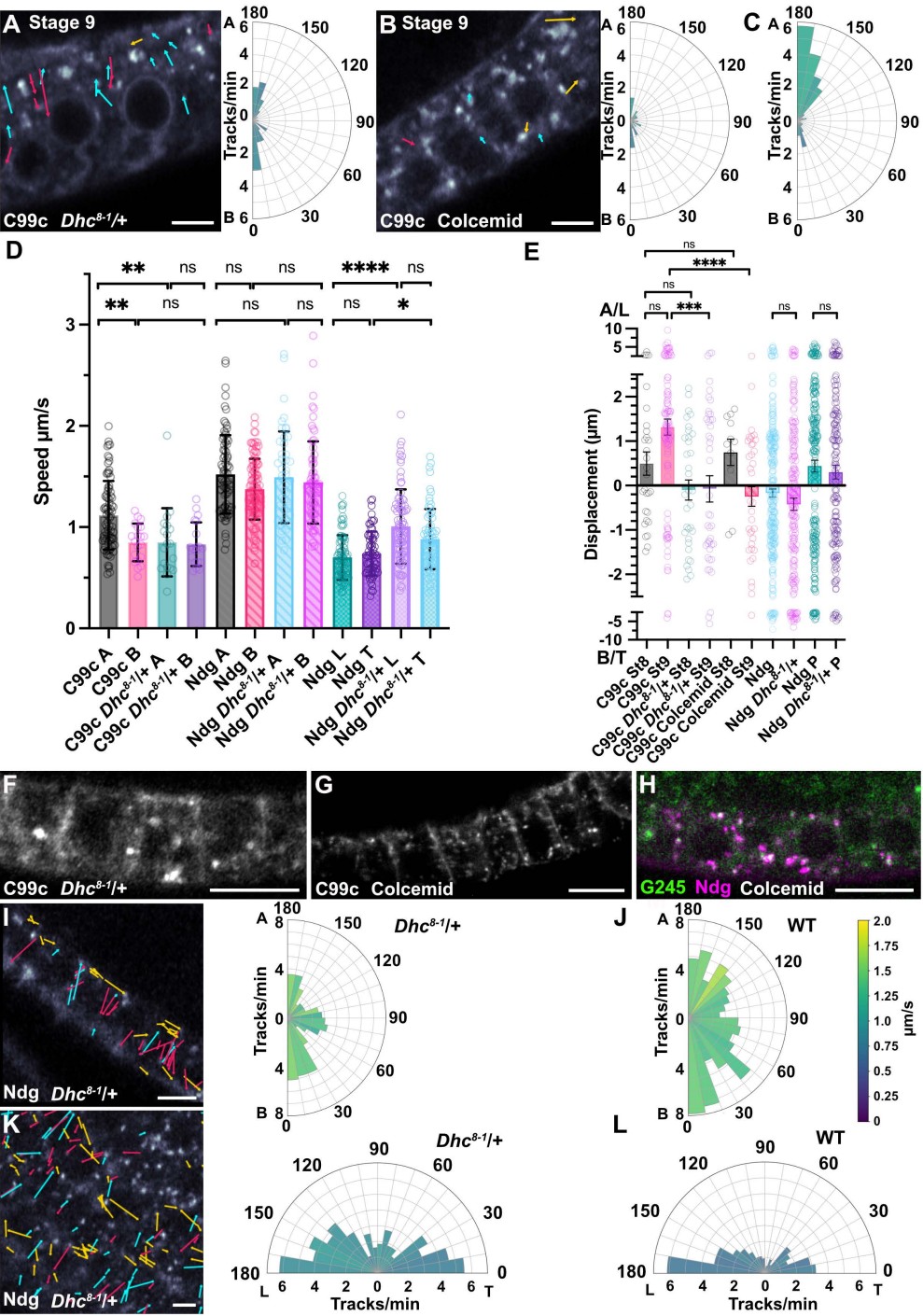

**Fig 8. The effects of a slow dynein mutant and microtubule depolymerisation on Cad99c and Ndg trafficking. A)** Tracks from a representative movie of Cad99c trafficking in a *Dhc*[8-1] heterozygous stage 9 egg chamber, 13 minutes after biotin addition. The polar plot on the right represents data from 3 Airyscan movies and shows the loss of the apical bias in Cad99c vesicle movements seen in WT egg chambers. Scale bar 5 μm. **B)** Tracks from a representative movie of Cad99c trafficking in a stage 9 egg chamber, 26 minutes after biotin addition that has been treated with colcemid to depolymerise microtubules. The polar plot on the right represents data from 4 movies. Scale bar 5 μm. **C)** A polar plot of Cad99c tracks in wild-type stage 9 egg chambers for comparison with (A) and (B). The plot represents data from 5 movies, and is the same as in Figure 5B. The scales in A) B) and C) are the same to allow for easy comparison. **D)** Graph showing the maximum curvilinear speeds (over a segment) of Cad99c and Ndg vesicles travelling apically or

basally in wild-type and $Dhc^{8-1}$/+ cells and for planar polarised Ndg tracks along the basal surface. A stands for Apical, B for Basal, L for Leading Edge, and T for Trailing Edge. Cad99c vesicles move significantly faster apically than basally, and significantly faster than apical tracks in the $Dhc^{8-1}$/+ mutant. The $Dhc^{8-1}$ mutation has no effect on the speeds of Ndg vesicle movements along the apical-basal axis, but increases the the speed of planar-polarised vesicle movements at the basal side of the cell. The bars and error bars represent the means and standard deviations and the circles represent individual tracks. Statistical significance was determined with a Kruskal-Wallis test using Dunn's multiple comparisons test, with significant differences between conditions indicated: * = p < 0.05, ** = p < 0.01, **** = p < 0.0001. **E)** Graph showing the mean displacement of Cad99c and Ndg tracks in wild-type and $Dhc^{8-1}$/+. Positive displacement is towards the apical side (A) of the cell, with negative values indicating displacement towards the basal side (B). The last two columns show the displacement of Ndg vesicles on the basal surface towards the leading edge of the cell (L). Bars and error bars represent the means and standard errors of the means (s.e.m), and circles represent individual tracks. The y axis is compressed at large displacement values to fit all the data in. Statistical significance for the C99c dataset was determined by a Kruskal-Wallis test using Dunn's multiple comparisons test, with significant differences between conditions indicated. Statistical significance for the Ndg pairs determined by Mann-Whitney test. * = p < 0.05, ** = p < 0.01, **** = p < 0.0001. **F)** An image of Cad99c RUSH in a $Dhc^{8-1}$/+ background, taken 43 min after biotin addition. Cad99c has been secreted to the lateral membrane, but not the apical membrane. Scale bar 10 μm. **G)** A projection of a Z-stack from a movie of Cad99c RUSH in a colcemid treated egg chamber 54 minutes after biotin addition. Cad99c has been delivered to the lateral membrane, but not to the apical membrane. Scale bar 10 μm. **H)** Image from a movie of Ndg trafficking in a colcemid-treated stage 8 egg chamber expressing GFP-G245. The image was taken 30 min after biotin addition and shows that most Ndg is still trapped in the Golgi ministacks. Scale bar 10 μm. **I)** Tracks from representative movie of Ndg trafficking in a stage 8 $Dhc^{8-1}$/+ egg chamber taken 24 minutes after biotin addition. Scale bar 5 μm. The polar plot on the right represents 3 movies and shows the number of tracks per minute. **J)** A polar plot showing the number of Ndg tracks per minute in WT stage 8 egg chambers for comparison with (I). **K)** Tracks from representative movie of Ndg trafficking at the basal side of a stage 8 $Dhc^{8-1}$/+ egg chamber taken 22 minutes after biotin addition. Scale bar 5 μm. The polar plot represents 2 movies and shows the number of Ndg tracks per minute. **L)** A polar plot showing the number of Ndg tracks per minute in wild-type stage 8 egg chambers for comparison with (K).

microtubule network by staining microtubules in WT and $Dhc^{8-1}$/+ egg chambers (S7J and S7K Fig). The net speed of apical movements at stage 9 was significantly reduced in the slow dynein mutant: in wildtype egg chambers, vesicles moved apically with a mean net speed of 1.1 μm/sec ± 0.34 (sd), whereas basally-directed vesicles moved more slowly at 0.85 μm/sec ± 0.19 (sd). In the $Dhc^{8-1}$/+ mutant, however, the apically-directed vesicles moved at the same speed as the basally-directed vesicles (0.85 μm/sec ± 0.34 (sd) and 0.83 μm/sec ± 0.22 (sd) respectively), which were unaffected by the mutation (Fig 8D). Clones of cells homozygous for $Dhc^{8-1}$ and $Dhc^{6-6}$, another slow Dynein mutant [66] showed similar reductions in the speed of apically-directed Cad99c vesicle movements and a delay in delivery of Cad99c to the apical membrane (S6D and S6E Fig). Clones of $Dhc^{6-6}$ showed no difference in the speeds of basally directed tracks compared to WT cells (S6D Fig, note: there were no $Dhc^{8-1}$ basal tracks to compare to the WT). Homozygous clones of a stronger allele, $Dhc^{4-16}$, produced gaps in the follicular epithelium and completely blocked Cad99c trafficking (S6F Fig). These observations confirm that Dynein is the major motor that transports Cad99C vesicles to the apical surface, whereas another, slower, motor mediates the basal movements.

Colcemid treatment leads to the depolymerization of almost all microtubules in the follicle cells, but a few stable microtubules persist in the apical region of the cells (Mock injected in S7A, S7C, S7E, and S7G Fig, colcemid treated in S7B, S7D, S7F, and S7H Fig). This leads to a dramatic reduction in the number of Cad99c vesicle movements, indicating that almost all Cad99c trafficking is microtubule-dependent (Figs 8B and S6B and S8 Movie). Depolymerizing microtubules with colcemid had an even more dramatic effect on where Cad99C is secreted, with 71% of movies showing exclusively lateral secretion or secretion laterally before apically (Fig 8G). This indicates that dynein-dependent apical transport along microtubules is the primary determinant of where Cad99c is secreted and implies that Cad99c-containing vesicles are competent to fuse with the lateral membrane. During the final stages of its secretion,

Cad99C normally accumulates in Rab11$^{+ve}$ recycling endosomes beneath the apical membrane (Fig 1D) and is then transported to the membrane by Myosin V [22]. Consistent with this, colcemid treatment leads to the partial relocalization of the Rab11$^{+ve}$ endosomes to the lateral side of the cell (S7I Fig).

In contrast to Cad99c, the movements of Nidogen-containing vesicles were unaffected by the $Dhc^{8-1}$ mutant, suggesting that their trafficking either does not require Dynein, or is mediated by a form of the Dynein motor complex whose speed is unaffected by the 8−1 mutation, which has not been molecularly characterized. (Fig 8D and 8I-8J, S9 Movie). Apically-directed Nidogen-containing vesicles move at the same speed as basally-directed ones in both wild-type cells and $Dhc^{8-1}$ mutant cells (Fig 8D). Furthermore, both move much faster than Cad99C vesicles, with an average speed of 1.5 μm/sec ± 0.4 (sd) apically and 1.4 μm/sec ± 0.3 (sd) basally. This is significantly faster than kinesin-1, suggesting that the basal transport is mediated by Khc73, a member of the KIF3 family of super-processive, fast plus end-directed motors that transports Col IV basally [20,67,68]. Since almost all microtubules grow from the apical surface with their plus ends extending basally, Khc73 is unlikely to be responsible for the fast apical movements of Ndg vesicles.

The planar polarized movements of Nidogen vesicles along the basal surface are significantly slower, with a mean speed of 0.7 μm/sec, in agreement with the proposal by Zajac and colleagues [20] that kinesin-1 is the predominant motor that transports vesicles along basal microtubules. Consistent with the requirement for kinesin-1 and Khc73 for Nidogen vesicle transport, depolymerizing microtubules with colcemid blocked Nidogen secretion (Fig 8H). Indeed, most Nidogen appeared to remain in the Golgi, as shown by its co-localization with Golgin245, suggesting that the exit of Nidogen from the TGN requires the pulling forces exerted by microtubule motor proteins. Finally, the movements of Nidogen vesicles along the basal side of the cell in both directions were significantly faster in $Dhc^{8-1}/+$ cells, although the bias in the direction of Ndg tracks was not affected (Fig 8D, 8K and 8L, S9 Movie). This suggests that dynein is not the motor protein responsible for transport towards the leading or trailing edges, although why these movements are faster in $Dhc^{8-1}/+$ is unclear.

## Discussion

Understanding how secretory cargoes are secreted to the correct domain of the plasma membrane requires being able to track and analyze vesicle movements at scale in a reliable and reproducible manner and MSP-tracker provides a powerful tool to do this. The CNN only requires small training sets and the same trained model can be used to analyze multiple experiments if the vesicles are similar in size and appearance. It is particularly good at tracking vesicle movements in noisy environments where other organelles are also visible in the same channel, such as the mini-Golgi stacks and the ER, in our RUSH experiments. The user-friendly interface of MSP-tracker also allows the results to be monitored at each step of the algorithm, facilitating the selection of the optimal parameters for vesicle selection and track linking. Testing the performance of MSP-tracker showed that it performs particularly well with real data, correctly classifying 90% of the ground truth vesicle trajectories identified by manual annotation. Two other leading tracking algorithms identified fewer than half of these ground truth trajectories, demonstrating that MSP-tracker is significantly better at analyzing vesicle movements in a noisy environment. The companion program, MSP-viewer allows users to review and correct the tracking results before analyzing the extracted trajectories. This feature significantly improves the resulting data, as it allows users to delete spurious trajectories, which are usually short and can often be removed simply by filtering by length. One can also add any trajectories that were missed by MSP-tracker, which are most often tracks that were not linked correctly. Once this straightforward manual curation is complete,

MSP-viewer can then be used to analyse the speeds and directions of the resulting trajectories, providing detailed information on the dynamics of vesicle movement. Both MSP-programs, extensive documentation, and instructional videos can be found at https://doi.org/10.17863/CAM.114339 and the software can be run on Windows, MacOS, or Unix machines. The dataset used to train and test MSP-tracker can be found here https://doi.org/10.17863/CAM.114338.2. One can also import tracks from other algorithms into MSP-viewer and use the simple, but powerful, interface to curate the tracks.

The trafficking of Cad99C to the apical surface of the follicle cells has been shown to depend on dynein and microtubules in studies in fixed samples [22]. Consistent with these results, we observe that Cad99c vesicles predominantly move towards the apical surface, particularly at stage 9, and that these movements are microtubule-dependent, as they largely cease after treatment with colcemid. Analysis of the vesicle trajectories reveals that they move faster and further towards the apical surface than they do basally, suggesting that different motors mediate their movements in each direction. Furthermore, two slow dynein mutants reduce the speed of apical movements without affecting the basal ones. These results demonstrate that dynein directly transports Cad99c vesicles toward the apical surface, whereas an unidentified, slower motor mediates the basal movements.

The observation that the slow Dynein mutant leads to the secretion of Cad99c at the lateral membrane of the follicle cells differs from the results of Khanal and colleagues (2018) [22], who found that knock-down of dynein led to the intracellular retention of Cad99c. This disparity is most probably due to the difference between removing most dynein activity by RNAi-mediated knock-down and just slowing the speed of the motor. Indeed, treating follicle cells with the dynein inhibitor Ciliobrevin prevents the exit of Cad99c from the Golgi, indicating that dynein is required for Cad99c trafficking to the plasma membrane (S10 Movie). By contrast, Cad99c can exit the Golgi in $Dhc^{8-1}/+$, but is no longer efficiently transported apically, allowing its lateral secretion.

The lateral secretion of Cad99c in $Dhc^{8-1}/+$ and colchicine-treated egg chambers indicates that the apical-basal organization of the microtubule cytoskeleton is the main determinant of where different cargoes are secreted. This is surprising since the fusion of vesicles containing apical cargoes with the apical membrane requires a specific t-SNARE protein in mammalian cells, Syntaxin 3, whereas the fusion of basolateral cargo vesicles requires Syntaxin 4 [69,70]. Drosophila lacks clear homologs of Syntaxin 3 and 4, and it is therefore possible that the apical and basolateral membranes contain the same fusion machinery, so that vesicles with apical cargoes are competent to fuse with the basolateral membrane and vice versa. In this case, the site of Cad99c secretion would be solely determined by its Dynein-dependent transport to microtubule minus ends at the apical membrane, explaining why it is mistargeted laterally when Dynein and microtubules are disrupted. Similarly, the secretion of Collagen IV at the basal part of the lateral membrane depends on its basal transport by Khc73 and kinesin-1, with knockdown of both motors leading to its secretion at the apical region of the lateral domain [20]. However, it is also possible that the mis-targeting of secretory cargoes is an indirect effect of disrupting microtubule-based transport, as t-SNARES also need to be delivered to the correct membrane. Microtubule disruption causes the mis-localization of Syntaxin 3 to the basolateral domain of MDCK cells, inducing the secretion of apical cargoes basolaterally [71]. Thus, there may be specific apical fusion factors in Drosophila whose localization also depends on Dynein and microtubules, and which lead to the lateral secretion of Cad99c when mislocalized, but the nature of these factors is unknown.

Knockdown studies have indicated that Khc73 is the main motor that transports ECM cargoes vesicles basally, while kinesin-1 may play a larger role in the planar polarized transport along the basal side of the cell [20]. Our results support this idea, since we observe that the

Nidogen-containing vesicles move towards the basal side at 1.4 µm/sec, consistent with transport by the fast kinesin-3 family member, Khc73, whereas the planar polarized movements across the basal surface occur at half this speed, as expected for kinesin-1. Nidogen vesicles show a clear bias in their planar-polarized basal movements towards the leading edge of the migrating follicle cells, indicating that most Nidogen is secreted from this side of the cell, at least when over-expressed with the RUSH system. Based on movies of Rab10[+ve] vesicle movements, it has been previously inferred that Collagen IV is secreted from the opposite, trailing side of the follicle cells [20]. However, the fact that Nidogen over-expression switches the direction of Rab10[+ve] vesicle movement indicates that Rab10 is not necessarily a reliable proxy for Collagen IV trafficking. We therefore directly determined the site of Collagen IV secretion by visualizing the endogenous protein accumulating laterally between expressing and nonexpressing follicle cells. These results confirm that Collagen IV is mainly secreted from the trailing edges of the cells. Thus, the follicle cells seem to have mechanisms to target the secretion of Nidogen and Collagen IV to opposite sides of the basal surface. Epithelial cells are generally thought to traffic secreted proteins to either the apical or basolateral plasma membrane, but the behavior of these ECM proteins demonstrates that they can target cargoes much more precisely.

Rab10 is required for the Khc73-dependent transport of ECM vesicles along microtubules to the basal side of the cell [20]. Once these Rab10[+ve] vesicles reach the basal side, it seems likely that other adaptor proteins take over from Rab10 to mediate the planar-polarized trafficking of Collagen IV and Nidogen vesicles in opposite directions. In this scenario, Rab10 would be a passenger on two distinct populations of basal vesicles, one containing Collagen IV that moves towards the trailing edge and another containing Nidogen that moves towards the leading edge. This would explain the shift in the direction of Rab10[+ve] vesicle movements on Nidogen over expression. The wave of Nidogen produced by the RUSH system will increase the population of Rab10[+ve] vesicles moving towards the leading edge, without affecting the population moving towards the trailing edge, and this will reverse the directional bias.

In conclusion, our results indicate that polarized secretion in epithelial cells is more complex than previously thought. MSP-tracker in combination with the RUSH system provides an efficient approach to visualize the trafficking of secretory cargoes in epithelial cells Combining these approaches with mutations and drugs that disrupt motors and trafficking factors may reveal the underlying logic of polarized secretion in epithelia.

## Materials and methods

### Imaging systems

Two microscopes were used to collect the RUSH tracking data for this study, a Zeiss 880 Airyscan microscope using the Airyscan "Fast" mode, using the 63x Water objective 1.2 NA, and a custom built spinning disc microscope [72].

The UAS-Rab10-YFP tracking data were collected on the Zeiss 880 Airyscan using Airyscan "Fast" mode with the 63x oil immersion objective 1.4 NA.

All other images were obtained using a Leica SP8 with white light laser, using the 63x oil objective, 1.4 NA.

For information on acquisition speeds for the tracking movies, microscope system used, Halo/SNAP dye used, and other parameters, see supplementary movie legends.

### Fly husbandry and lines

Fly lines used in this study are described in **Table 1**.

For RUSH experiments, flies were raised at 25°C until pupation, at which point they were moved to 18°C to reduce leakiness of the RUSH system.

**Table 1. Fly lines used in this study.**

| Line | | Location | Reference | Notes |
|---|---|---|---|---|
| 1 | 2xUAST-SA-KDEL/CyO | attP40 and attP51C1 | [28] | |
| 2 | UAST-SBP-Halo-Cad99c | attP2 | This Study | |
| 3 | UAST-SBP-SNAP-Ndg | attP2 | This Study | |
| 4 | UAST-SBP-SNAP-Ndg | 86Fb | This Study | |
| 5 | TrafficJam-Gal4/CyO (P{GawB}NP1624) | 2L | [75] | |
| 6 | GFP-Sec16 | Endogenous locus (X) | This Study | |
| 7 | GFP-Golgin-245 | Endogenous locus (2R) | [76] | |
| 8 | GFP-GMAP (Gmap$^{CC00492}$) | Protein trap (X) | [77] | |
| 9 | YFP-Rab11/TM6B | Endogenous locus (3R) | [78] | |
| 10 | w;; Dhc64c$^{8-1}$/TM6B | 3L | [63] | |
| 11 | w; 2xUAST-SA-KDEL/CyO; UAST-SBP-Halo-Cad99c/TM6B | | | Combination of 1 and 2 |
| 12 | w; 2xUAST-SA-KDEL/CyO; UAST-SBP-SNAP-Ndg/TM6B | | | Combination of 1 and 3 |
| 13 | w; 2xUAST-SA-KDEL/CyO; UAST-SBP-Halo-Cad99c, UAST-SBP-SNAP-Ndg/TM6B | | | Recombination of 2 and 4, combined with 1 |
| 14 | GFP-Sec16; TrafficJam-Gal4/CyO | | | Combination of 5 and 6 |
| 15 | w; TrafficJamGal4, GFP-Golgin245/CyO | | | Recombination of 5 and 7 |
| 16 | GFP-GMAP; TrafficJam-Gal4/CyO | | | Combination of 5 and 8 |
| 17 | w; TrafficJamGal4/CyO; YFP-Rab11/TM6B | | | Combination of 5 and 9 |
| 18 | w; TrafficJam-Gal4/CyO; Dhc64c[8−1]/TM6B | | | Combination of 5 and 10 |
| 19 | y, w; P{UASp-YFP-Rab10}21 | 3L | RRID:BDSC_9789 | |
| 20 | y1 w*; P{PTT-un}vkgG00454, FRT40A/ CyO | 2L | RRID:BDSC_98343 recombined with FRT 40A | Recombination of Vgk-GFP trap line and FRT40A |
| 21 | hsFlp; P{w[+mC] = Ubi-mRFP.nls}2L P{ry[+t7.2] = neoFRT}40A/CyO | 2L | hsFlp added to RRID:BDSC_34500 | |
| 22 | UAST-SBP-Halo-Cad99c | 86Fb | This study | |
| 23 | y, w, hs-Flp;; ubi-GFP, FRT2A/ TM3, Sb | 3 | | |
| 24 | w;; ubi-GFP, FRT2A, UAST-SBP-Halo-Cad99c/ TM6B | 3 | | Recombination of 22 and 23 |
| 25 | w; 2xUAST SA-KDEL/ CyO; ubi-GFP, FRT2A, UAST-SBP-Halo-Cad99c/ TM6B | 2 and 3 | | Combination of 1 and 24 |
| 26 | w;; Dhc$^{6-6}$, FRT2A/ TM6B | 3 | [63] | Allele recombined with FRT2A |
| 27 | w;; Dhc$^{8-1}$, FRT2A/ TM6B | 3 | [63] | Allele recombined with FRT2A |
| 28 | w;; Dhc$^{4-16}$, FRT2A/ TM6B | 3 | [63] | Allele recombined with FRT2A |
| 29 | y, w, hsFlp; TrafficJam Gal4/Cyo; MKRS/ TM6B | 1, 2 and 3 | | |
| 30 | y, w, hsFlp; TrafficJam Gal4/Cyo; Dhc$^{6-6}$, FRT2A/ TM6B | 1, 2 and 3 | | Combination of 26 and 29 |
| 31 | y, w, hsFlp; TrafficJam Gal4/Cyo; Dhc$^{8-1}$, FRT2A/ TM6B | 1, 2 and 3 | | Combination of 27 and 29 |
| 32 | y, w, hsFlp; TrafficJam Gal4/Cyo; Dhc$^{4-16}$, FRT2A/ TM6B | 1, 2 and 3 | | Combination of 28 and 29 |

Flies heterozygous for *Dhc64c$^{8-1}$* show reduced trafficking speeds of *bicoid* mRNA particles [65], and we therefore used this mutant to test if the speeds of Cad99c and Ndg vesicles movements were affected by this allele.

UAST-SBP-Halo-Cad99c was created by cloning the signal peptide of the Cad99c followed by a GSGSGSG linker, the sequence for streptavidin binding peptide (SBP), Pro-Ala-Gly, then the HaloTag sequence, another GSGSGSG linker, and the rest of Cad99c into pUAST(AttB) using NEB HiFi assembly. UAST-SBP-SNAP-Ndg was created by cloning the signal peptide of cg25c followed by the SBP sequence, then a Pro-Ala-Gly linker, then the SNAP sequence, a GSHMRSRPTS linker and the Ndg sequence into pUAST(AttB) using NEB HiFi assembly. The cargoes were designed to ensure that the SBP was facing the lumen of the ER and therefore could access the streptavidin-KDEL hook. Transgenic flies were made by injecting into the listed landing sites.

Sec16 was endogenously tagged using CRISPR/Cas-mediated homology-directed repair [73,74]. A pCFD3-gRNA-Sec16 plasmid and a donor construct were injected together into TH_attP2 embryos at a concentration of 75ng/μl and 100ng/μl, respectively. To make the donor construct, EGFP along with a GSGSGSG linker and ∼1,000bp long homology arms were cloned into pBluescriptSK+ using NEB HiFi Assembly. The endogenous Sec16 ATG was removed, and to prevent Cas9 from recutting the repaired DNA,the PAM site was mutated. Injected flies and their progeny were screened for the presence of the GFP sequence and multiple independent stable lines were recovered.

## Sample preparation for live imaging

Flies were fattened on yeast for 2 days at 18° C. Ovarioles were dissected out of ovaries and the muscle sheath in Schneider's insect medium supplemented with insulin (7.5 μg/ml, Merck I9278). For SNAP-Ndg flies, ovarioles were initially labeled in Schneider's plus 0.5 μM SNAP-SurfaceBlock for 5–10 min to block nonegg-chamber-autonomous Ndg labelling of the ECM. Ovarioles were labeled in Schneider's plus 1 μM SNAP or HaloTag ligands for 15–20 min at room temperature before being transferred to an 8-well poly-lysine coated μ-slide (Ibidi) in 100 μl Schneider's medium. To induce release of the cargo from the ER, 100 μl Schneider's containing 800 μM D-Biotin (Merck) was added to the medium, to give a final concentration of 400 μM D-Biotin.

For colcemid treated ovarioles, the abdomens of flies were injected with 0.4 mg/ml colcemid in Schneider's medium 1 hour before dissection. Labeling and cargo release were carried out as above, but 0.4 mg/ml colcemid was present in all the solutions. Colcemid stocks were dissolved in DMSO, as the usual carrier ethanol caused independent effects on the egg chambers, while DMSO did not.

## Correlation analysis for cargoes with CellMask, GFP-Sec16, and GFP-Golgin245

Image analysis was carried out using Fiji [54]. Pearson's correlation coefficients were calculated for the cargo and marker (e.g., Cell Mask, GFP-Sec16, GFP-GMAP, GFP-G245) signal in each frame of the time series using a custom script (https://github.com/gurdon-institute/Egg_Chamber_Analysis/blob/main/Pearson_frames.py).

For Cell Mask, GFP-GMAP, GFP-G245, and UAS-Rab10-YFP, the zero time point was set to a value of zero, and the difference between other timepoints and the zero time point was calculated for each movie. For GFP-Sec16, as there is significant overlap at the zero time point, we set the endpoint of the movie to a value of zero, and calculated the difference between the other timepoints and the endpoint for each movie.

The changes in correlation were then entered into GraphPad Prism, and each dataset transformed to be on a scale of 0–1 for graphing purposes. The transformation was $Y = Y/K$, where $K$ = the mean maximum change in correlation. We were not able to use the raw correlation values, as the brightness of SNAP/Halo and CellMask labeling varies from egg chamber to egg chamber, whereas the change in correlation is consistent between samples.

## Quantifying Nidogen secretion with WGA (Wheat Germ Agglutinin)

We were unable to use the Pearson's correlation for WGA as the staining changed both in brightness and localization throughout the movie (i.e., at the beginning of the movie, the basal surface of the egg chamber was labeled, but by the end, it was brighter, and the lateral and apical membranes were also labeled)., We therefore quantified the basal signal of Nidogen using the basal WGA signal to define the basal region.

Egg chamber basal signal was quantified using a custom script (https://github.com/gurdon-institute/Egg_Chamber_Analysis/blob/main/Jenny_Basal_MeasureROICheck1_5basalR.py) to identify egg chambers and measure a 1.5 µm inner band containing the basal membranes of follicle cells. In order to robustly map egg chamber ROIs, the script first finds the largest WGA signal region in each frame, and then uses the convex hull if its area is greater than twice that of the original ROI. Egg chambers in each frame were then filtered to use the mapping from the previous frame if there was an overlap of less than 0.75. This gives reliable, roughly ellipsoid ROIs in frames where signal intensity varies, and a continuous boundary is not detected.

The output of the script includes a measurement of the mean brightness value in the Basal ROI. This was used to calculate the change in brightness over time, with time zero set to a value of zero. The values were then entered into GraphPad Prism, and each dataset was again transformed to be on a scale of 0–1 for graphing purposes. The transformation was $Y = Y/K$, where K = the mean maximum change in brightness.

## Fixed RUSH stainings of egg chambers

RUSH was carried out in egg chambers, co-expressing both SBP-SNAP-Ndg and SBP-Halo-C99c. Egg chambers were dissected in Schneider's as usual and incubated in 400 µM biotin for 5 min (to enable RUSH to proceed to the ERES and *cis*-Golgi) before being fixed in 4% formaldehyde in PBS for 20 min with rotation. After fixation, samples were washed 3x 5 min in PBS, then incubated with Halo-JF549 ligand (Promega) and SNAP-SiR (NEB) for 1 hour at 37°C with shaking at 500 rpm. Samples were washed 6x 10 min in PBT (0.1% Triton), and then mounted in Vectashield (Vectalabs) for imaging.

## Antibody stainings of RUSH egg chambers

For antibody stainings of egg chambers, ovaries of fattened flies were dissected into Schneider's medium supplemented with insulin. Ovarioles were incubated in Schneider's medium plus 400 µM biotin for 0, 15, or 30 min before being transferred to 4% formaldehyde in PBS and fixed for 20 min with rotation. Ovarioles were washed 3x5min in PBS, then incubated

**Table 2.** Number of movies used for cargo marker correlation analysis.

| Cargo → | Cad99c | Ndg |
|---|---|---|
| Marker ↓ | | |
| Sec16 | 10 | 11 |
| GMAP | 8 | 15 |
| G245 | 11 | 12 |
| CellMask | 9 | N/A |
| WGA | N/A | 9 |
| Rab11 | 18 | N/A |
| Rab10 | N/A | 16 |

with Halo-TMR ligand (Promega) for 1 hour at 37°C with shaking at 500 rpm. Samples were washed 3 × 5 min in PBS, then blocked in PBT (0.2% Tween20) with 10% BSA overnight at 4°C. Ovarioles were then incubated for 48 hours at 4°C in PBT 1% BSA with 1:2000 Rabbit anti-Rab11 [79] and 1:1,000 Goat anti-Rabbit. Ovarioles were washed 4x30min in PBT 1% BSA, then stained overnight at 4°C with 1:500 Donkey anti-Rabbit Alexa488 and Donkey anti-Goat Alexa647 secondary antibodies in PBT 0.1% BSA. Samples were washed 3x10 min in PBT and mounted in Vectashield plus DAPI (Vectorlabs). Images were taken on the Leica SP8 WLL with the 63x oil objective. 1.4 NA.

## Microtubule stainings of egg chambers

For microtubule stainings of egg chambers, fattened flies were either mock injected with Schneider's medium, or with 0.4 mg/ml colcemid in Schneider's medium, 1 hour before dissection. Ovaries were dissected into either Schneider's medium supplemented with insulin, or also containing 0.4 mg/ml colcemid. Ovarioles were incubated for 30 min before being transferred to 10% formaldehyde in PBT (2% Tween20) and fixed for 10 min. Ovarioles were washed 3 × 5 min in PBT (2% Tween20), then blocked in PBT (0.2% Tween20) with 10% BSA for 2 hours. Ovarioles were then incubated overnight at 4°C in PBT (0.2% Tween20), 1% BSA and a 1:200 of FITC-conjugated anti-α-tubulin antibody (F2168, Merck, clone DM1A). Ovarioles were then washed 6 × 10min in PBT (0.2% Tween20) and mounted in Vectashield plus DAPI (Vectorlabs). Images were taken on the Leica SP8 WLL with the 63× oil objective. 1.4 NA.

## Staining and quantification of extracellular Collagen IV at leading and trailing cell edges

hsFlp/ +; P{PTT-un}vkgG00454, FRT40A/ nls-RFP, FRT40A females were collected and fattened on yeast (with males present), and subjected to 2–3 x 2 hour 37°C heatshocks at 12 hour intervals. Alternatively, pupae were heatshocked for 3 days with 2 hour 37°C heatshock at 12 hour intervals, and then adults fattened for 24–48 hours at 25°C with yeast.

Ovaries were dissected out in Schneider's supplemented with insulin, and the ovarioles removed from the muscle sheath. Ovarioles were transferred to a drop of Schneider's with 1:50 dilution of Atto647n GFP-nanobody (Proteintech) to label the extracellular Collagen for 20 min. Ovarioles were transferred to fresh Schneider's and imaged as previously on an 8-well poly-lysine coated μ-slide (Ibidi). Timelapse Z-stacks were taken to allow the direction of rotation to be determined. The amount of extracellular collagen at the leading and trailing edges of the cells was then measured as follows. In FIJI[48] a segmented line was drawn along the leading or trailing edges of a patch of cells. This was converted to an area, the area expanded to a band of ~0.2um wide, and the integrated density was measured. The integrated density was divided by the number of cells in the patch and used to compare Collagen levels at the leading and trailing edges of the cells. Nonboundary cells (i.e., those that contacted Vkg-GFP expressing cells on both sides) were measured as a control.

## Using MSP-tracker and MSP-viewer to track Cad99c, Ndg, and Rab10 particles

MSP-tracker was used to find particle tracks following the method described in the user manual. MSP-viewer was used to assess the tracks found by MSP-tracker. Tracks were filtered by length to remove short (<700 nm) tracks. Remaining tracks were assessed on a track-by-track basis, and tracks were added manually, extended, edited, or deleted as necessary. Once tracks had been curated, they were filtered to remove anything of less than 1,000 nm long, and data

were exported for further analysis. Excel was used to filter data into tracks going in different orientations, and Graphpad Prism was used to represent the data.

### Tracking in Dhc64c mutant clones

hsFlp/+; TrafficJam Gal4/ 2x UAST-SA-KDEL; ubi-GFP, FRT2A, UAST-SBP-Halo-Cad99c/ *Dhc\**, FRT2A females were collected and fattened on yeast (with males present), and subjected to 2–3 x 2 hour 37°C heatshock at 12 hour intervals. Alternatively, pupae were heatshocked for 3 days with 2 hour 37°C heatshock at 12 hour intervals, and then adults fattened for 48–72 hours at 18°C with yeast. Halo labeling and RUSH imaging were done as for the other live RUSH movies, but with a still image taken in two channels (to mark the clones) and tracking movies taken in a single channel, to enable fast imaging.

### Western blotting

For each sample, three ovaries were homogenized in 25 μl Schneider's medium and 25 μl 2× Laemmli sample buffer and boiled at 95°C. 10 μl of the sample were loaded per lane on a 4%–12% Bis-Tris polyacrylamide gel. The membrane was probed for BiP and Tubulin using rabbit anti-GRP78 (BiP) (StressMarq biosciences SPC-180S) at 1:1,000, and mouse anti-tubulin (DM1A) at 1:1,000. Goat anti-mouse Alexa 680 and goat anti-rabbit Alexa790 were used as secondary antibodies at a dilution of 1:10,000. The blot was imaged using a LiCor Odyssey system.

### The tracking algorithm

The particle tracking algorithm of the MSP-framework consists of particle detection and frame-to-frame object association (linking) (Fig 3B). A pre-processing step enhances the appearance of the particles and improves the detection results. The background intensity is estimated from m consecutive frames (parameters can be set manually) and subtracted from the original frame. Finally, Contrast Limited Adaptive Histogram Equalization is applied to each frame to enhance the contrast.

The particle detection exploits an extended multiscale version of the spot-enhancing filter (SEF). The SEF highlights spots by convolving the original image $g(x, y)$ with a Laplacian-of-Gaussian kernel $LoG(x, y, \sigma)$. In the MSSEF [52], a set of SEF is applied sequentially with an intensity-based threshold after each iteration. This way, each iteration can be represented by Equation 1.

$$f\left(x, y, \sigma^k\right) = LoG\left(x, y, \sigma^k\right) * \left(b\left(x, y, \sigma^{k-1}\right) g\left(x, y\right)\right), \tag{1}$$

where $\sigma^k$ is a standard deviation of the LoG, and $b(x, y, \sigma^{k-1})$ is a binary mask extracted from the previous iteration. The threshold for each iteration $T^k$ is defined by the mean intensity of the image $\mu^k$ and its standard deviation $\varsigma^k$ weighted by a user-defined value $c$, Equation 2.

$$T^k = \mu^k + c\varsigma^k, \tag{2}$$

The MSSEF exposes spots of different sizes and intensities and provides a reduction of the false detection rate related to the noise levels. The particle candidates are extracted based on the local maxima of the final MSSEF image. The coordinates of the candidates are used to define a region of interest (ROI) for each candidate. The default ROI size is 16×16 pixels, but the value can be adjusted for larger particle sizes. The CNN network classifies the ROIs to provide the final detection results. Training on small ROIs instead of an entire image allows one

to use a light-weight CNN and apply the same learned model for different imaging modalities provided that the appearance and size of the particles are similar.

The light-weight CNN architecture consists of two convolutional layers followed by max pooling and two fully-connected layers. The softmax activation function is used for the output layer, and Relu activation for the rest. The network is trained with binary cross-entropy loss function and root mean square propagation optimizer. The training data is augmented with rotation and flipping.

Finally, the locations of the selected candidates are refined to provide subpixel localization using the iterative center-of-mass refinement for the task [42].

The data association of the detected particles is solved with a two-step linking approach. The first step of the data association considers the distance between detections and existing tracks, solving the assignment problem sequentially for each frame with the Hungarian algorithm [53]. The weight is represented by the Euclidean distance between the detection and already existing tracks. The formed tracklets are used to build longer trajectories.

A Bayesian network (BN) is used to compute the connectivity score for each pair of the tracklets. This probabilistic graphical model is used to represent a set of discrete variables and their conditional dependencies and is implemented using pgmpy library[51,80]. The score expresses the probability of these tracklets being linked based on their characteristics. Each variable (node) of the BN represents a single characteristic of the tracklet pair. These characteristics can be divided into four main groups: particle appearance, motion, location, and temporal position.

The particle appearance is represented by the intensity node. It indicates the similarity in average intensity between the ROIs in the last frame of the first tracklet and the first frame of the second tracklet. We expect the intensity of the particle to change along the trajectory but have similar values in succeeding frames. The motion is defined by two nodes: orientation and speed, where the similarity in orientation and speed of particle movement is defined. The

**Table 3. Quantitative trajectory measures provided by the MSP-viewer.**

| Measures | Formulation |
|---|---|
| Total distance traveled | $d_{total} = \pounds_{i=1}^{N-1} d\left(p_i + p_{p+1}\right)$ |
| Net distance traveled | $d_{net} = d\left(p_i + p_N\right)$ |
| Maximum distance traveled | $d_{max} = \max_i\left(d\left(p_1 + p_i\right)\right)$ |
| Total trajectory time | $t_{traj} = (N-1)t$ |
| Net orientation | $\alpha$ |
| Mean brightness (normalized) | $b = \dfrac{1}{b_{max}N} \pounds_{p=1}^{M} b_p$ |
| Mean curvilinear speed | $\nu = \dfrac{1}{N-1} \pounds_{i=1}^{N-1} \dfrac{d\left(p_i + p_{i+1}\right)}{t}$ |
| Mean straight-line speed | $\nu_{line} = \dfrac{d_{net}}{t_{traj}}$ |
| Max curvilinear speed over a segment | $\max\left(\nu_1, \nu_2, \dots \nu_w\right)$ |
| Number of moving segments | $k$ |
| Average moving segment time | $t_{avg} = \dfrac{\pounds_{j=1}^{k} t_j}{k}$ |

$p_i$, position of the $i^{th}$ particle; N, number of detections in the trajectory; t, trajectory time; M- number of pixels in the region of interest; $b_p$, pixel brightness; $b_{max}$, value of the brightest pixel in the image sequence.

location of the tracklets is defined based on the Euclidean distance between the coordinates of the last detection in the first tracklet and the first detection in the second tracklet. The temporal position is defined by sequence and gap variables. The sequence node characterizes the successiveness of the first and second tracklets. This node is defined by two parent nodes: an order and an overlap. The gap node represents a temporal gap between the last detection in the first tracklet and the first detection in the second tracklet. The conditional probability table defines the contribution of each parent node. The introduced connectivity score is a binary variable with True, meaning that the pair of tracklets are connected and belong to the same trajectory. The inference over the BN provides the probability of the connectivity score (CS) to be True, $P_c$:

$$Pc = P\left(CS = True | I, O_r, S_p, O_d, O_v, G\right), \tag{3}$$

where $I$ is a difference in intensities, $O_r$ – in orientation, $S_p$ - in speed, $O_d$ – order of the tracklets, $O_v$ – presence of the overlap, and $G$ is a temporal gap between the tracklets. When all the possible tracklet pairs are evaluated, the assignment problem is addressed with the Hungarian algorithm [53], and the tracklets are assembled into the tracks.

Parameters for detection and linking can be set in MSP-tracker GUI before running the tracking algorithm for the entire timelapse sequence.

### Trajectory evaluation in track-viewer

The individual trajectory window of the MSP-viewer provides several quantitative trajectory measures (Table 3). Traveled distance is described by the total, net and maximum distances. The total distance is the entire trajectory length. In contrast, net distance represents the length between the first and the last points, and the maximum distance is defined by the largest interval between any two points of the trajectory. The total trajectory time is the number of frames scaled by the frame rate. The net orientation is defined by the first and last trajectory's points. The curvilinear speed is calculated from the total distance and can be defined as the mean of the instantaneous speeds, while the net distance defines the straight-line speed.

The speed is evaluated for the entire trajectory (average speed), for the trajectory segments where the particle is moving (moving speed), and the maximum curvilinear speed is calculated for a given time interval by a sliding window over the moving segments. To allow segmentation of short trajectories, where the traditional Mean Square Displacement (MSD) methods are not applicable, we exploit a 1D U-Net segmentation [81]. The method provides segmentation of the directed motion of the particle from the remaining trajectory. The software provides both options, MSD and 1D U-Net based approaches.

### Registration of the image sequence

Some image sequences have an undesirable movement of the egg chamber in the frame. We have registered the sequences before using the tracking software to prevent the movement's influence on the trajectory evaluation. We have used our custom Python code and the SimpleITK library [82], using mutual information between two neighboring frames to register image sequences.

### Supporting information

**S1 Fig. (relating to Fig 1) – RUSH does not cause ER stress.** BiP/GRP78 expression is increased when the ER experiences stress, so we measured expression in egg chambers expressing the Cad99c and Ndg RUSH constructs, with or without the 2x Streptavidin-KDEL

insertions that retain the cargoes in the ER. Tubulin was used as a loading control. BiP/GRP78 levels do not change when the 2x Streptavidin-KDEL hook was expressed to retain the cargoes in the ER, indicating that trapping the cargoes does not cause ER stress.
(TIFF)

**S2 Fig. (relating to Fig 2) – Time course of Cad99c and Nidogen secretion. A)** Stills from RUSH movies of SBP-Halo-Cadherin 99c trafficking in the follicle cells after release from the ER. Cad99c is shown in an early stage 9 egg chamber. Time indicates minutes since the addition of biotin. Cad99c is labeled with Halo-OregonGreen, At 35 min a small amount of lateral membrane signal can now be seen, which is more pronounced at 45 min. Scale bars 10 μm. **B,C&D)** Still images taken at different time points are biotin addition from a RUSH movie of SBP-Halo-Cadherin 99c (magenta) trafficking in cells expressing B) GFP-GMAP (*cis*-Golgi), C) GFP-Golgin245 (*trans*-Golgi and TGN) and D) YFP-Rab11 (TGN and RE) as secretory pathway markers. Scale bars 5 μm. **E&F)** Stills from a RUSH movie of SBP-SNAP-Nidogen trafficking in cells expressing E) GFP-GMAP, or F) GFP-Golgin245. Scale bars 5 μm.
(TIFF)

**S3 Fig. (relating to Fig 2) – The specificity of ER exit sites in the follicular epithelium. A)** Fixed images of an egg chamber expressing both SBP-SNAP-Ndg and SBP-Halo-Cad99c in the RUSH system. Egg chambers were fixed after 5 min in biotin, to allow RUSH to proceed as far as the ER exit sites and *cis*-Golgi. SNAP-Ndg was labeled with SNAP-SiR dye, and Halo-C99c with Halo-JF549. ER exit sites were marked by the presence of GFP-Sec16. **B)** shows a zoomed in view of the yellow boxed area in A) showing that both Ndg and Cad99c are present at all the ERES that are labeled with GFP-Sec16 (endogenously tagged). Scale bars 10 um.
(TIFF)

**S4 Fig. (relating to Fig 5)—Spinning disc imaging of Cad99c trafficking. A)** Tracks from representative movie taken on a spinning disc microscope of Cad99c trafficking in a stage 9 egg chamber, 30 min after biotin addition. The polar plot on the right shows the strong apical bias in the direction of the tracks. Movies were taken at 4 fps using a custom-built spinning disc system. **B)** Tracks from a representative movie taken on a spinning disc microscope of Cad99c trafficking in a stage 9 egg chamber, 35 min after biotin addition. The polar plot on the right represents data from 2 spinning disc movies and shows that the strong apical bias has disappeared, as the exocytosis of C99c is now balanced by endocytosis. Track data can be found in S4 Data. Movies used for tracking can be found at https://doi.org/10.17863/CAM.114338.2.
(TIFF)

**S5 Fig. (relating to Fig 6)—Spinning disc imaging of Nidogen trafficking. A)** Tracks from representative movie taken on a spinning disc microscope of Ndg vesicle movements along the apical-basal axis of a stage 8 egg chamber. The polar plot on the right represents data from 2 spinning disc movies, and shows a slight basal bias in Ndg movements. **B)** Tracks from a representative movie taken on a spinning disc microscope of Ndg vesicle movements Ndg tracks in a transverse section through the middle section of the follicle cells (see diagram). The polar plot represents data from 3 spinning disc movies and shows no clear bias in track direction. **C)** Tracks from a representative movie taken on a spinning disc microscope of Ndg vesicle movements in a transverse section at the basal side of the follicle cells (see diagram). The leading and trailing (L and T) directions of follicle cell migration are indicated in the polar plot on the right, which shows that most tracks are directed towards the leading edge. Track data can be found in S4 Data. Movies were taken at 2 fps. Movies used for tracking can be found at https://doi.org/10.17863/CAM.114338.2.
(TIFF)

**S6 Fig. (relating to Fig 8)—*Dhc*$^{8-1}$/+, colcemid treatments, and Dynein mutant clones. A)** Tracks from a representative movie of Cad99c trafficking in a *Dhc*$^{8-1}$ heterozygous stage 8 egg chamber 33 min after biotin addition. Scale bar 5 μm. The polar plot on the right represents data from 3 movies. **B)** Tracks from a representative movie of Cad99c trafficking 24 min after biotin addition in a stage 8 egg chamber that has been treated with colcemid to depolymerize microtubules. Scale bar 5 μm. The polar plot on the right represents data from 4 movies **C)** Polar plot representing data from 2 movies showing the number of Cad99c tracks per minute in wild-type stage 8 egg chambers. Graph scales in A) B) and C) are the same to allow for easy comparison. **D)** Graph showing the maximum curvilinear speeds (over a segment) of Cad99c vesicles traveling apically in wild-type and *Dhc* mutant clones. Cad99c vesicles move significantly faster in wild-type cells than in either the *Dhc*$^{6-6}$ or *Dhc*$^{8-1}$ homozygous cells. The bars and error bars represent the means and standard deviations and the circles represent individual tracks. Statistical significance was determined with a two-tailed Welch's *T* test with significant differences between conditions indicated: * = p < 0.05. **E)** A still image of Cad99c RUSH in a *Dhc*$^{6-6}$ homozygous clone marked by the loss of nuclear GFP (green) 51 min after biotin addition. Cad99c has progressed further through the secretory pathway in the wildtype cells (green nuclei) than in the mutant cells (arrowheads). Scale bar 5 μm. **F)** A still image of RUSH in *Dhc*$^{4-16}$ homozygous clones marked by the loss of nuclear GFP (green) at 26 min after biotin addition. *Dhc*$^{4-16}$ is a strong allele of *Dhc64c* and has caused gaps to form in the epithelium. Cad99c has reached the plasma membrane in the wild-type cells (GFP positive nuclei), whereas it has not reached the membrane in the mutant cells (asterisks). Scale bar 5 μm. Track data can be found in S3 Data. Movies were taken at 4.7 fps. Movies used for tracking can be found at https://doi.org/10.17863/CAM.114338.2.
(TIFF)

**S7 Fig. (relating to Fig 8) – Rab11 and microtubules in *Dhc*$^{8-1}$/+, colcemid treatments, and Dynein mutant clones. A), C), E), G)** β-tubulin staining in mock injected flies. Scale bar 10 μm. **B), D), F), H)** β-tubulin staining in flies injected with colcemid. These images were collected with the same microscope settings and laser power colour scale as the mock injected images. Scale bar 10 μm. **B'), D'), F'), H')** have all been adjusted to show the details of the remaining microtubules. **C) and D)** show microtubules in cuboidal follicle cells. **G) and H)** show microtubules in columnar follicle cells. **I)** Rab11 and Cad99c localization in colcemid-treated egg chambers 30 min after the addition of biotin. Rab11 positive recycling endosomes are no longer enriched apically and can be seen laterally and basally with Cad99c. Scale bar 5 μm. **J)** β-tubulin staining in yw flies. Scale bar 10 μm. **K)** β-tubulin staining in *Dhc*$^{8-1}$/+ flies. Scale bar 10 μm.
(TIFF)

**S1 Table. The number of vesicles traveling in each direction for all tracked RUSH movies of Cad99c and Ndg.** $\chi^2$ tests were performed to determine if the tracks were directionally biased. The null hypothesis was that equal numbers of tracks would travel apically/basally, or leading/trailing.
(XLSX)

**S1 Raw Images. The uncropped, unedited version on the blot shown in S1 Fig.**
(PDF)

**S1 Data. Data used to generate the graphs in Figs 2G and 2H, and 7B.**
(XLSX)

**S2 Data. Data used to generate the graphs in Fig 4A and 4C.**
(XLSX)

**S3 Data. Tracking data from Airyscan movies that were used to generate Figs 5, 6, 7D-F, 8A-E and 8I-L, and S6A-D.**
(XLSX)

**S4 Data. Tracking data from Spinning disc movies that were used to generate S4 and S5 Fig.**
(XLSX)

**S5 Data. Viking-GFP localization data from clonal egg chambers that were used to generate the graph in Fig 7H.**
(XLSX)

**S1 Movie. Associated with Fig 2A.** Cadherin99c RUSH – Images taken every 5 min. SBP-Halo-C99c labeled with Halo-OregonGreen. Time represents time since biotin addition. Cadherin99c can be seen trafficking through the secretory pathway in an early stage 9 egg chamber. The large puncta that can be seen 10 min after cargo release are the Golgi ministacks. The small puncta seen subapically in the cells from 15 min onward correspond to Rab11 positive recycling endosomes, through which Cad99c travels.
(AVI)

**S2 Movie Associated with Fig 2B.** Nidogen RUSH – Images taken every 5 min. SBP-SNAP-Ndg labeled with SNAP-SiR. Time represents time since biotin addition. Nidogen can be seen trafficking through the secretory pathway in a stage 8 egg chamber. The large puncta seen at 10 min after cargo release are the Golgi ministacks.
(AVI)

**S3 Movie. Associated with Figs 2 and S2.** Cadherin99c RUSH with secretory pathway markers. Images taken every 1 min. SBP-Halo-C99c labeled with Halo-JF646, except when combined with CellMask DeepRed (then labeled with Halo-OregonGreen). Time represents time since biotin addition. The movies are ordered to follow the route through the secretory pathway. The first section shows Cad99c partially colocalizing with Sec16 (ER exit sites) en route to the Golgi, the second shows colocalization with GMAP (*cis*-Golgi), and the third with Golgin245 (*trans*-Golgi). The final section shows the accumulation of Cad99c at the apical membrane, labeled by CellMask.
(AVI)

**S4 Movie. Associated with Figs 2 and S2.** Nidogen RUSH with secretory pathway markers. Images taken every 1 min. SBP-SNAP-Ndg labeled with SNAP-SiR. Time represents time since biotin addition. The movies are ordered to follow the route through the secretory pathway. The first section shows Nidogen partially colocalizing with Sec16 (ER exit sites) en route to the Golgi, the second shows colocalization with GMAP (*cis*-Golgi), and the third with Golgin245 (*trans*-Golgi). The final section shows the accumulation of Nidogen in the basement membrane, labeled by Alexa488-Wheatgerm agglutinin.
(AVI)

**S5 Movie. Associated with Fig 5.** Cadherin99c RUSH – All movies were taken on the Zeiss Airyscan using Airyscan fast mode at 4.5 frames per second. SBP-Halo-Cad99c was labeled with Halo-JF549. Time represents time since biotin addition. Vesicle tracks > 1 μm are shown overlaid on the righthand side of the movie. Part 1: stage 8, 21 min after addition of biotin; vesicles move in a generally apical direction. Part 2: stage 9, 29 min after addition of biotin; almost all Cad99c move apically. Part 3: stage 9, 39 min after addition of biotin; vesicles can

be seen moving basally as well as apically, as Cad99c that has reached the apical membrane is being endocytosed at this point.
(AVI)

**S6 Movie. Associated with Fig 6.** Nidogen RUSH – All movies taken on the Zeiss Airyscan using Airyscan fast mode. SBP-SNAP-Nidogen labeled with SNAP-SiR. Time represents time since biotin addition. Vesicle tracks >1 μm are shown overlaid on the righthand side of the movie. Part 1: stage 8, 23 min post-biotin. Taken at 3.8fps on the Airyscan. Cross section through follicle cells. Vesicles move rapidly both apically and basally. Part 2: stage 8, 40 min after biotin addition. Taken at 2.3 frames per second. Section through the middle of the follicle cells. Vesicle tracks are not planar polarized. Part 3: stage 8, 33 min after biotin addition taken at 2.3 frames per second. Section through the basal region of the follicle cells. Tracks are mostly planar polarized along the axis of egg chamber rotation.
(AVI)

**S7 Movie. Associated with Fig 7.** Nidogen RUSH with UAS-Rab10-YFP. All sections are through the basal region of the follicle cells. SBP-SNAP-Nidogen labeled with SNAP-SiR. Part 1: stage 8 – taken on Leica SP8 at 1 frame per minute. Time represents time since biotin addition. Around 11 min after the addition of biotin, some Nidogen starts to colocalize with the Rab10 puncta, with peak colocalization around 20 min after the addition of biotin. Parts 2–4: all stage 8 and taken on the Zeiss Airyscan with Airyscan fast mode at 1.59 frames per second. Part 2 shows Rab10 (Green) and Nidogen (Magenta). Yellow arrowheads indicate examples of Rab10 positive vesicles; cyan arrowheads indicate examples of Ndg and Rab10 positive vesicles. There are plenty of Rab10 vesicles that do not contain Nidogen, but most Nidogen vesicles are labeled with Rab10 for at least part of the track. Part 3: Rab10 channel only. Part 4: Nidogen channel only. Part 5: UAS-Rab10-YFP expressed in the absence of UAS-SBP-SNAP-Nidogen. Stage 8 and taken on the Zeiss Airyscan with Airyscan fast mode at 1.56 frames per second.
(MP4)

**S8 Movie. Associated with Figs 8 and S5.** Cadherin99c RUSH in $Dhc^{8-1}$/+ and after colcemid treatment. All movies were taken on the Zeiss Airyscan using Airyscan fast mode at 4.5 frames per second. SBP-Halo-C99c was labeled with Halo-JF549. Time represents time since biotin addition. Tracks > 1 μm are shown overlaid on the right-hand side of the movie. Part 1: Cadherin99c RUSH in $Dhc^{8-1}$/+ at stage 9. There are many fewer tracks than in wild-type cells, and the apical bias is lost. Part 2: Cadherin99c RUSH in $Dhc^{8-1}$/+ at stage 8. Part 3: Cadherin99c RUSH after Colcemid treatment at stage 9. There are far fewer tracks than in nontreated egg chambers, and those that are present are less polarized. Part 4: Cadherin 99c RUSH after Colcemid treatment at stage8. There are far fewer tracks than in nontreated egg chambers.
(AVI)

**S9 Movie. Nidogen RUSH with slow Dynein.** All movies were taken on the Zeiss airyscan using Airyscan fast mode. SBP-SNAP-Ndg was labeled with SNAP-SiR. Tracks > 1 μm are shown overlaid on the righthand side of the movie. Part 1: Nidogen in $Dhc^{8-1}$/+ at stage 8. Taken at 3.8 frames per second. Cross-section through the follicle cells. Nidogen trafficking is similar to wild-type, with lots of tracks moving apically and basally .Part 2: Nidogen in $Dhc^{8-1}$/+ at stage 8. Taken at 2.3 frames per second. Section through the basal side of the follicle cells. Tracks are still planar-polarized along the rotational axis.
(MP4)

**S10 Movie. Cadherin99c RUSH with ciliobrevin D treatment.** Images taken every 5 min. SBP-Halo-C99c labeled with Halo-JF646. Time represents time since biotin addition.

Ciliobrevin D was added to inhibit dynein function at 10 min post-biotin. After Ciliobrevin D treatment, Cadherin99c remains trapped in the Golgi ministacks.
(AVI)

## Acknowledgments

We thank Akira Nakamura for providing us with the Rab11 antibody, the Bloomington and Kyoto Stock Centres for fly lines, Franck Perez for introducing us to the RUSH system and the St Johnston lab for help and advice. We would like to acknowledge the Gurdon Institute Imaging Facility for help with microscopy and image analysis. We are particularly grateful to John Overton for his help in generating all the transgenic lines used in this study.

## Author contributions

**Conceptualization:** Jennifer H. Richens, Helen L. Zenner, Jens Rittscher, Daniel St Johnston.

**Data curation:** Jennifer H. Richens, Mariia Dmitrieva.

**Formal analysis:** Jennifer H. Richens, Mariia Dmitrieva.

**Funding acquisition:** Jens Rittscher, Daniel St Johnston.

**Investigation:** Jennifer H. Richens, Helen L. Zenner.

**Methodology:** Jennifer H. Richens, Mariia Dmitrieva, Helen L. Zenner, Nadine Muschalik, Jens Rittscher.

**Project administration:** Daniel St Johnston.

**Resources:** Helen L. Zenner, Nadine Muschalik, Jade Glashauser, Carolina Camelo, Stefan Luschnig, Sean Munro.

**Software:** Mariia Dmitrieva, Richard Butler, Jens Rittscher.

**Supervision:** Daniel St Johnston.

**Validation:** Jennifer H. Richens, Mariia Dmitrieva.

**Visualization:** Jennifer H. Richens, Mariia Dmitrieva, Daniel St Johnston.

**Writing – original draft:** Daniel St Johnston.

**Writing – review & editing:** Jennifer H. Richens, Mariia Dmitrieva, Sean Munro.

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
