## [Editor Report · Decision Letter 0]

8 May 2024

Dear Dr St Johnston,

Thank you for submitting your manuscript entitled "Tracking exocytic vesicle movements reveals the spatial control of secretion in epithelial cells." for consideration as a Research Article by PLOS Biology. Please accept my sincere apologies for the long delay in getting back to you as we consulted with an academic editor about your submission.

Your manuscript has now been evaluated by the PLOS Biology editorial staff, as well as by an academic editor with relevant expertise, and I am writing to let you know that we would like to send your submission out for external peer review.

Once your full submission is complete, your paper will undergo a series of checks in preparation for peer review. After your manuscript has passed the checks it will be sent out for review. To provide the metadata for your submission, please Login to Editorial Manager (https://www.editorialmanager.com/pbiology) within two working days, i.e. by May 10 2024 11:59PM.

Kind regards,

Richard

Richard Hodge, PhD

rhodge@plos.org

PLOS

---

## [Decision Letter · Decision Letter 1]

20 Jun 2024

Dear Dr St Johnston,

Thank you for your patience while your manuscript "Tracking exocytic vesicle movements reveals the spatial control of secretion in epithelial cells." was peer-reviewed at PLOS Biology. It has now been evaluated by the PLOS Biology editors, an Academic Editor with relevant expertise, and by several independent reviewers.

In light of the reviews, which you will find at the end of this email, we would like to invite you to revise the work to thoroughly address the reviewers' reports.

As you will see below, overall the reviewers were interested in the topic of the paper, and commented on the mix between methods and biological insights. However, they also had some serious concerns, particularly about the documentation and benchmarking of the MSP-tracker. Furthermore, following discussions with the Academic Editor, we feel that this study would work better as a 'Methods and Resources' paper. To that end, it will be essential that you bolster the MSP-tracker, to provide more benchmarking, to make it more user friendly, and to provide a test dataset and more documentation for this approach. In addition, we also think that all biologically focused reviewer comments should be addressed to support the conclusions of the study

Given the extent of revision needed, we cannot make a decision about publication until we have seen the revised manuscript and your response to the reviewers' comments. Your revised manuscript is likely to be sent for further evaluation by all or a subset of the reviewers.

**IMPORTANT - SUBMITTING YOUR REVISION**

*Re-submission Checklist*

*Published Peer Review*

*PLOS Data Policy*

*Blot and Gel Data Policy*

Sincerely,

Suzanne

Suzanne De Bruijn, PhD,

Associate Editor

PLOS Biology

sbruijn@plos.org

REVIEWS:

Reviewer #1: In the manuscript by Richens et al., the authors present a significant advancement in understanding the targeting of specific secretory cargoes to distinct domains of the plasma membrane in epithelial cells. Using the classic model system of Drosophila epithelial follicle cells, they apply two innovative methods: the RUSH system and the MSP tracking framework. While I appreciate these groundbreaking methods, I have some major concerns about certain conclusions and hope the authors can address them in their revised manuscript.

(1) 1) The MSP tracker is a powerful method, particularly effective for tracking in noisy and dim images due to its machine learning module. However, while following their detailed manual with step-by-step instructions, I encountered persistent error messages in Anaconda when running tests to identify particles, specifically showing "AttributeError: 'str' object has no attribute 'decode'." I speculate that this issue may stem from differences in string function definitions across Python versions (I am using Python 3.11.7 on a Windows 10 computer). Due to the complexity and length of the original code, I was unable to fully review the quality of the MSP tracker. I highly recommend that the authors provide a step-by-step video tutorial to facilitate broader adoption and usage of the MSP tracker.

(2) The authors used a slower dynein mutant (Dhc[8-1]) to demonstrate that Cad99c apical transport is dependent on dynein. In Dhc[8-1]/+ heterozygous cells, the apical velocity of Cad99c decreased from 1.1 μm/sec to 0.85 μm/sec. I have two concerns regarding this part: (a) Cytoplasmic dynein is known to affect microtubule organization in both dividing cells (mitotic spindle formation) and non-dividing cells (neurons, nurse cells, etc.). Therefore, the authors need to ensure that there is no significant microtubule reorganization in Dhc[8-1]/+ cells; (b) Since the measurement was done in Dhc[8-1]/+ cells, it is difficult to determine whether dynein or another unidentified motor is responsible for the residual apical transport. I recommend that the authors measure Cad99c transport in Dhc[8-1] homozygous FRT clones. A previous paper from the same lab (Reference [58], Vítor Trovisco et al., 2016, eLife) reported the use of homozygous mutant germline clones of Dhc[8-1], indicating that this dynein mutant does not cause significant cell division defects and thus should allow the production of FRT clones in follicle cells. Alternatively, a trans-heterozygous mutant of Dhc[6-10]/Dhc[8-1] or TRiP RNAi lines targeting Dhc64C could be used.

(3) While the role of dynein in transporting Cad99c vesicles to the apical membrane is clear, the section on Ndg transport is less so, and some conclusions are puzzling.

(a) The authors showed that overexpression of Ndg shifts Rab11 directionality towards the leading edge, whereas in wild-type cells, Rab11 is predominantly directed towards the trailing edge. This suggests that Ndg and Collagen IV (and probably other vesicles toward the trailing edge) compete for Rab11 vesicles. This raises the question of whether endogenous Ndg shows a preference for the leading edge. I recommend the authors conduct a similar FRT clone experiment (as was done for the Vkg clone) to determine if endogenously secreted Ndg preferentially goes to the leading edge.

(b) It is puzzling that the authors suggest dynein is not required for the apical transport of Ndg vesicles. Given that the microtubule network is highly polarized, with minus ends near the apical side and plus ends extending towards the basal side, do the authors suggest another minus-end-directed motor is involved? Alternatively, it is possible that Ndg requires less active dynein for apical transport, making the effect not significant in Dhc[8-1] heterozygous cells. The authors further suggest that Khc-73 is required for the basal transport of Ndg vesicles. This hypothesis could be tested by combining the Khc-73 mutant with the RUSH method and MSP-tracker.

(c) The authors argue that "transport towards both the leading and trailing edges is mediated by plus-end-directed motors antagonized by dynein." However, microtubules on the planar plane are highly polarized, extending from the leading edge to the trailing edge with their minus ends at the trailing edge. I find it difficult to understand how plus-end-directed motors could mediate movement in both directions along such a polarized network. Based on velocity and previous publications, the authors suggest kinesin-1 is responsible for these movements, although the planar Ndg velocity (~0.7 μm/sec) is on the higher end of the reported kinesin-1 velocity (~0.4 μm/sec). The involvement of kinesin-1 could be tested using well-established kinesin-1 mutants. Additionally, at the end of the results section, the authors suggest that "vesicle movements towards the leading edge are significantly faster than those towards the trailing edge in Dhc[8-1]/+, suggesting that different motors move the vesicles in each direction." However, Figure 8D indicates not significant ("ns") between the last two sets of data."

Minor revision points:

1. For supplemental figures, the figure panels are not referred to sequentially in the text. Additionally, they should be renamed as "Supplement for Figure X" (as in the figure legend) instead of "Figure Sx" (as in the text). Otherwise, it is confusing that only Figure S1, S2, S5, S6, and S8 are mentioned.

2. The overexpression patterns of Cad99c and Ndg were not shown. On page 4, Figure 1C is referred to incorrectly: "although some protein leaks into the lateral domain because of overexpression (Figure 1C)". Figure 1C does not match this text description. Later on page 5, Figure 1C is referred to again (this time correctly): "On the addition of biotin, it is released and most Cad99c is localized in the Golgi after 15 minutes (Figure 1C)". If possible, please provide antibody staining to show the endogenous localization of Cad99c and Ndg for readers to compare.

3. On page 5: "Expressing two copies of UAS-Streptavidin-KDEL is sufficient to retain all SBP-Halo-Cad99C in the ER without activating the ER stress response, as monitored by the levels of BiP [39] (Figure S1)." Throughout the manuscript, BiP was never properly introduced. Please include a brief introduction to BiP.

4. On page 5, regarding Rab11 recycling endosomes: "Cad99c co-localizes with Rab11 in puncta immediately below the apical membrane before it reaches the apical surface, indicating that it passes through Rab11-positive recycling endosomes en route to the cell surface (Figure 1D)." Does a dominant negative Rab11 mutant affect Cad99c reaching the apical microvilli?

5. For Figures 2G and 2H, what are the sample sizes for each quantification? SEM or 95% CI are needed on these curves to show the range for each time point.

6. Do the follicle cells rotate in Movie S6? If not, how are the leading edge versus trailing edge defined?

7. On page 11: "This increase in directionality correlates with the more highly polarized organization of the microtubules along the apical-basal axis of columnar follicle cells compared to the cuboidal cells at stage 8 (Figure S8G and K)." It is difficult to determine which example has more polarized MTs. Some quantitative measurement is needed here. Alternatively, the authors could use a more direct trackable marker, such as EB1, to show a more polarized MT network at stage 9.

8. In Figure 7G, it is hard to discern whether there is a bias of extracellular Viking-GFP accumulation at the trailing edge. Please add arrowheads to make it clearer where to look.

9. On page 14: "Colcemid treatment leads to the depolymerization of almost all microtubules in the follicle cells, but a few stable microtubules persist in the apical region of the cells (Figure S8)." Figure S8 is large and complex. It should be more specific; please refer to Figure S8F, H, J, L.

10. On page 15, at the beginning of the discussion: "[20] The trafficking of Cad99C to the apical surface of the follicle cells has been shown to depend on dynein and microtubules in studies in fixed samples [52]." It is unclear why reference [20] is placed at the beginning of the discussion.

Reviewer #2: This manuscript by Richens et al. describes a study of polarized secretory traffic in the Drosophila follicular epithelium. The researchers employ the RUSH technology, previously implemented for in vivo/in animal studies in Drosophila (Glashauser et al., 2023), to create waves of two secretory cargos and study in live recordings their progress through the secretory pathway towards the plasma membrane. The first of these two experimental cargos is an apically localized transmembrane adhesion protein (Cad99C), the second one a basally localized ECM protein (Nidogen). For the quantitative analysis of the live recordings of the two cargos, the manuscript describes the development of a tracking tool called MSP-tracker that allows them to quantitate cis-trans progress with respect to secretory pathway markers, as well as direction and speed within the cell. Based on these data, Richens et al show that post-Golgi movement of each cargo displays a directional bias: apical for Cad99C, basal for Nidogen. In addition, after analysis of dynein mutants and colcemid treatments, the authors make points of some novelty on how the two cargos are moved within the cell and targeted to their place of secretion: in the case of Cad99C, dynein mutation and colcemid treatment abolish apical bias (in number of apically-directed tracks and their higher speed), which results in basolateral secretion; basally biased movement of Nidogen, in contrast, is independent of dynein, but, interestingly, its whole post-Golgi movement depends on microtubules, as colcemid abolishes it completely. A major point of interest extracted from the data is that Nidogen basal secretion is planarly polarized towards the leading edge of the migrating follicle cells, which is in contrast to Collagen IV, another ECM protein, previously shown to be targeted through kinesin transport to the opposite, trailing end (Zajac and Horne-Badovinac, 2022).

I support publication of this manuscript provided several points are addressed. These hopefully do not involve extensive new experiments, but additional analysis of existing data, discussion, re-interpretation and rewriting.

1. A point central to the spatial control of secretion the manuscript is trying to address is whether ERES are all the same or specific in what they secrete. Could the authors extract from their movies the answer to this question by comparing the spatial distribution of the ERES-Golgi origins of the two cargos?

2. The authors state that Cad99C is secreted apically through Rab11 endosomes. This was also a conclusion in Khanal et al., 2016. Indeed, Rab11 in Fig 1D shows colocalization with Cad99C in cytoplasm and both are enriched sub-apically. However, Rab11 resides in most Golgi, as reported recently in other Drosophila cell types (Zhou et al., 2023). Therefore, it would be important to know whether Rab11 and Cad99C track together from trans-Golgi to apical PM. The actual movie (Movie 1 does not show Rab11) and a colocalization plot like the one for Rab10 in Fig 7B could be shown to evaluate alternatives: first, that Rab11 and apically secreted Cad99C follow the same trans-Golgi to PM route but in different carriers; second, that transmembrane Cad99C is not secreted but just recycled through Rab11 once secreted.

3. In Fig. 7D-F it is shown that a Nidogen wave shifts Rab10 distribution bias from trailing to leading. This is difficult to understand considering that Collagen IV, secreted to trailing edge, is also transported in Rab10 vesicles. How would the motors or the microtubules know what the Rab10 vesicle contains? Does the Nidogen wave cause Collagen IV mistargeting to leading edge or are they sorted into different Rab10+ populations? Is it possible that the wave itself (not the fact that it's Ndg) somehow causes the shift? Would a collagen wave do the same (change direction to leading)? I understand the experiments to address these questions may be difficult to design, but the title points to this differential planar targeting as the main experimental finding in the paper. At least, the authors should check that endogenous Ndg (protein trap, not UAS or UAS wave) is secreted towards the leading edge in normal conditions.

4. The authors should comment on the fact that Cad99C is basolaterally mistargeted upon microtubule cytoskeleton disruption, which is in contrast with Khanal et al., 2016, who reported basolateral accumulation of unsecreted Cad99C (endogenous). The authors state that a low level of basolateral leakage for UAS-SBP-Cad99C already occurs without disrupting microtubules (Fig 1C, 2nd paragraph of results), opening the possibility that mistargeting would not occur with the endogenous protein or at endogenous levels. This is an important point because it is used to argue that directionality of microtubule-based transport is enough to explain apical targeting (sufficiency of directionality to explain basal targeting was already demonstrated by the kinesin mutations that mistarget collagen IV to the apical side in Zajac 2022).

Minor points:

A. Fig 2G,H: It may be worth pointing out that the Cad99C transmembrane cargo seems to take longer to traverse the Golgi than soluble Ndg. The authors could discuss this in the context of intra-Golgi transport models and compare with speed figures in previous literature.

B. In Fig 2G,H, Ndg takes longer to traffic from the trans-Golgi to the PM than Cad99C. Is this explained by the differential apical and basal microtubule-based routes the two cargos follow? Their different speeds or displacement lengths? Again, perhaps this is in the data already.

C. It is stated that Collagen IV, because it is a dimer (actually, an obligate heterotrimer of the two Collagen IV chains), is not amenable to the RUSH approach. I wonder if this is based on speculation or actual attempts. I see in principle no problem if the hook is located in a place on either chain that does not affect trimerization. The other chain could be imaged instead of the hooked/overexpressed one to avoid imaging a mixture of trimer and excess monomer.

D. The fact that UAS-SBP-Ndg localizes to basement membrane should be shown (data not shown).

E. I did not understand the experiment in 7 G,H. Was it in doubt that Collagen IV is secreted to the trailing edge of cells (Zajac 2022)?

F. The fact that colcemid completely inhibits Ndg transport perhaps deserves some discussion.

Reviewer #3: In this article, Richens et al. investigate the tracking of Cadherin 99C and Nidogen during their secretion using the RUSH system. They achieve this by 1) developing a new tracking tool (MSP-tracker and viewer) and 2) employing the tool to study the secretion of RUSH cargoes. This manuscript is an interesting mix of image analysis development and biological experiments. My two main feedback on this manuscript are:

The biological observations remain limited at this stage. What is tracked is also unclear (see below). Clarification on the tracking and imaging conditions (3D and speed) is essential, and I cannot provide any recommendations without them.

The tracking tool developed could be better documented.

The methods section is poorly documented, making the data hard to evaluate. A more detailed and comprehensive account of the data would be beneficial.

The authors could consider whether Methods & Resources Articles would be a more fitting format for communicating their findings. As the work can be clearly divided into tool development and biological discoveries, I will provide feedback on these two categories.

Analyses of cargo secretion:

The authors appear to claim that they successfully tracked individual vesicles using 2D imaging with a confocal microscope, capturing one frame every 5 minutes. However, tracking vesicles in the cell volume under these conditions is impossible. Fast 3D imaging (on a second-time scale) and 3D tracking would be required to track individual vesicles. The authors need to clarify how these experiments were performed and analyzed.

The supplementary videos are pixelated and show a large drift. In addition, there are no video examples of successful tracking of individual vesicles after registration. Without convincing evidence of successful analyses, it is impossible to evaluate the biological findings further.

MSP-tracker and viewer:

Kudos to the authors for sharing their code. The GitHub repo has minimal documentation, though, and only a few users will be able to use this tool out of the box. I would not recommend that my lab use this tool at this stage.

I could not find a test dataset associated with the software. Including a test dataset would allow users to better understand and evaluate the tool.

A video showing the software in action would be beneficial for users to grasp its functionality and application.

In the manuscript, the rationale for developing a new tracking tool for this particular application could be better explained. If the main issue is signal-to-noise, why not include denoising in the analysis pipeline?

While the authors effectively assess their software on synthetic and real data, the comparison with other tracking tools (TrackMate and SpotTracking) lacks any methodology. Given the numerous options available in TrackMate and SpotTracking, it is crucial to provide information on how these comparisons were conducted. In addition, a comprehensive parameter sweep is necessary for a fair comparison.

The author indicates, "MSP-tracker performed much better than the other programs on this metric, as it detected nearly 90% of the annotated tracks compared to 40% for Trackmate and SpotTracking (Figure 4C)". But isn't the MSP tracker trained to detect the correct spots using manual annotations? What detectors were used in TrackMate? For instance, would training a Stardist/ilastik model trained to detect the spots perform as well? Is the issue in the tracking or the detection of the spots to start with?

The manuscript lacks references to the vast ecosystem of other available tracking tools, especially other tracking tools evolving in a similar space (i.e., tools available in Python)

The manuscript lacks information on the limitations of the tool and possible future developments. Is 3D tracking possible?

Reviewer #4: The authors studied trafficking of vesicles to examine polarized secretion in epithelial cells using the RUSH system [14, 15]. A vesicle tracking software, MSP-tracker and MSP-viewer, was developed to quantify the dynamics of trafficking from the generated time-lapse movies.

MSP-tracker is an extended version of the authors previous work [45] and uses deep learning. Vesicles are detected by the multi scale spot enhancing filter (MSSEF) [46], and then a CNN classifier is employed for candidate vesicle selection to improve the detection efficiency. Pretrained models for the CNN classifier are available in the MSP-tracker, and the classifier can also be trained on new data. Tracking is performed by a two-step linking approach using tracklets and a connectivity score computed by a Bayesian network (BN). The assignment problem is solved by the Hungarian algorithm. From the trajectories, different measures are computed to quantify trafficking (e.g., distance, total time, orientation).

The developed tracking method was evaluated using vesicle data from the Particle Tracking Challenge (CTC) [Chenouard 2014] and real data of vesicles from the RUSH system. For the CTC data, the performance falls within the mid/upper range of published results. For the vesicle data from the RUSH system, the method outperforms two other software, Trackmate [23,32] and SpotTracking [49].

The developed interactive tracking software was applied to quantify trafficking of post-Golgi vesicles in time-lapse movies from the RUSH system. The results reveal that Cadherin 99C vesicles move apically which suggests that microtubule organization plays an important role in the secretion process. It also turns out that Nidogen vesicles undergo planar-polarised movement.

The manuscript is well written, the developed software outperforms other software, and relevant biological findings were obtained. I suggest revising the manuscript and addressing the following points:

1. More information on the architecture of the used Bayesian network (BN) should be provided.

2. In the literature review on object tracking the authors mention the references [17-27,30-32]. More recent deep learning methods for particle tracking also applied to CTC vesicle data should be included.

3. Fig 4, "coloured areas refer to the range of the results": More information should be given how "range of the results" is defined.

4. Fig 4, "MSP-tracker performs extremely well for the Jaccard similarity coefficient for the entire tracks, JSCθ": The term "extremely well" should be relaxed.

5. "[20]The trafficking" and "[Chenouard2014]" should be corrected.

---

## [Decision Letter · Decision Letter 2]

13 Feb 2025

Dear Dr St Johnston,

Thank you for your patience while we considered your revised manuscript "Tracking exocytic vesicle movements reveals the spatial control of secretion in epithelial cells." for publication as a Research Article at PLOS Biology. Please accept my sincere apologies for the delays that you have experienced during this round of the peer review process. This revised version of your manuscript has been evaluated by the PLOS Biology editors, the Academic Editor and three of the original reviewers. Please note that Reviewer #1 has provided their comments in a word document which you can find attached to this e-mail.

Based on the reviews, I am pleased to say that we are likely to accept this manuscript for publication, provided you satisfactorily address the remaining points raised by the reviewers. Please also make sure to address the following data and other policy-related requests that I have provided below (A-G):

(A) After discussions with the editorial team and given some of the reviewer comments in the initial round of review, we think that your manuscript would be a better fit as a ‘Methods and Resources’ article at the journal. We do appreciate the hybrid Research/Method nature of the paper given the insights provided into the spatial control of secretory pathways in epithelial cells, but we feel that the strong methodological advance provided by the MSP-tracker tool should be highlighted. I would be grateful if you could please tick ‘Methods and Resources’ as the article type in the online submission form upon resubmission (if this option is unavailable then I can also do this on your behalf).

(B) In light of point A, we routinely suggest changes to titles to ensure maximum accessibility for a broad, non-specialist readership. In this case, we would suggest the following edit to the title. Please ensure you change both the manuscript file and the online submission system, as they need to match for final acceptance:

“MSP-tracker: a versatile vesicle tracking software tool used to reveal the spatial control of polarized secretion in Drosophila epithelial cells”

(C) You may be aware of the PLOS Data Policy, which requires that all data be made available without restriction: http://journals.plos.org/plosbiology/s/data-availability. For more information, please also see this editorial: http://dx.doi.org/10.1371/journal.pbio.1001797

-Supplementary files (e.g., excel). Please ensure that all data files are uploaded as 'Supporting Information' and are invariably referred to (in the manuscript, figure legends, and the Description field when uploading your files) using the following format verbatim: S1 Data, S2 Data, etc. Multiple panels of a single or even several figures can be included as multiple sheets in one excel file that is saved using exactly the following convention: S1_Data.xlsx (using an underscore).

-Deposition in a publicly available repository. Please also provide the accession code or a reviewer link so that we may view your data before publication.

Figure 2G-H, 4C, 5A-D, 6A-D, 7B, 7D-F, 7H, 8A-E, 8J, 8L, S3A-B, S4A-C, S5A-B

(D) Please also ensure that each of the relevant figure legends in your manuscript include information on *WHERE THE UNDERLYING DATA CAN BE FOUND*, and ensure your supplemental data file/s has a legend.

(E) We require the original, uncropped and minimally adjusted images supporting all blot and gel results reported in the following Figures:

Figure S1

We will require these files before a manuscript can be accepted so please prepare and upload them now. Please carefully read our guidelines for how to prepare and upload this data: https://journals.plos.org/plosbiology/s/figures#loc-blot-and-gel-reporting-requirements

(F) Please note that per journal policy, we do not allow the mention of "data not shown", "personal communication", "manuscript in preparation" or other references to data that is not publicly available or contained within this manuscript (page 17). Please either remove mention of these data or provide figures presenting the results and the data underlying the figure(s).

(G) Please ensure that your Data Statement in the submission system accurately describes where your data can be found and is in final format, as it will be published as written there.

We expect to receive your revised manuscript within 1 month.

*Published Peer Review History*

*Press*

Best regards,

Richard

Richard Hodge, PhD

rhodge@plos.org

Reviewer remarks:

Reviewer #1: See the attached review

Reviewer #2: In this revised version of the manuscript, Richens et al. have addressed satisfactorily most of the points I raised in my review. I only have two remaining comments:

- Point 1: I disagree that this is a tangential issue and strongly recommend that this result be included in the manuscript. Consider the alternative where Cad99C and Ndg are released from different ERES: in that case, where the two different ERES populations locate, within the cell and with respect to microtubule tracks, would need to be assessed before concluding differential post-Golgi trafficking. Showing that both cargos traffic from all ERES makes the strongest possible case that they are actively sorted and differentially transported post-Golgi. In addition, it is an important finding that two such different cargos (TM and soluble ECM) are equally collected in all ERES. It raises the biological significance of the paper.

- Point 3 (also point 3 in referee #1's review): It is unfortunate that the direction of endogenous Ndg secretion could not be determined. This would have strengthened the claim that Ndg and Col IV follow divergent basal routes. The technical reasons for this are understandable, but I would ask that the possibility that endogenous and overexpressed Ndg behave differently is acknowledged.

Reviewer #4: Most of my comments were addressed.

Ref 43 should be corrected ("DPTAT of P in FMIUDLR")

---

## [Editor Report · Decision Letter 3]

5 Mar 2025

Dear Dr St Johnston,

On behalf of my colleagues and the Academic Editor, Anna Akhmanova, I am pleased to say that we can accept your manuscript for publication, provided you address any remaining formatting and reporting issues. These will be detailed in an email you should receive within 2-3 business days from our colleagues in the journal operations team; no action is required from you until then. Please note that we will not be able to formally accept your manuscript and schedule it for publication until you have completed any requested changes.

PRESS

Best wishes, 

Richard

Richard Hodge, PhD

rhodge@plos.org

PLOS
